# Contextual inference through flexible integration of environmental features and behavioural outcomes

Jessica Passlack[1,2]*, Andrew F. MacAskill[1]*

**1** Department of Neuroscience, Physiology and Pharmacology, University College London, London, United Kingdom, **2** Centre for Discovery Brain Sciences, University of Edinburgh, Edinburgh, United Kingdom

* jpasslac@ed.ac.uk (JP); a.macaskill@ucl.ac.uk (AFM)

## Abstract

The ability to use context to flexibly adjust our decision-making is vital for navigating a complex world. To do this, the brain must both use environmental features and behavioural outcomes to distinguish between different, often hidden contexts; and also learn how to use these inferred contexts to guide behaviour. However, how these two interacting processes can be performed simultaneously remains unclear. Within the brain it is thought that interaction between the prefrontal cortex (PFC) and hippocampus (HPC) supports contextual inference. We show that models using environmental features (similar to those proposed to be implemented in hippocampus) readily support context-specific behaviour, but struggle to differentiate ambiguous contexts during learning. In contrast, models using behavioural outcomes (similar to those proposed in PFC) can stably differentiate contexts during periods of learning, but struggle to guide context-specific behaviour. We show that supporting feature-based with outcome-based strategies during learning overcomes the limitations of both approaches, allowing for the formation of distinct contextual representations that support contextual inference. Moreover, agents using this joint approach reproduce both behavioural- and cellular-level phenomena associated with the interaction between PFC and HPC. Together, these results provide insight into how the brain uses contextual information to guide flexible behaviour.

### Author summary

To behave flexibly, animals must determine which situation or context they are in, even when the information that defines that context is incomplete or no longer visible. This problem, known as contextual inference, is particularly challenging when different situations share similar sensory features and must be distinguished based on past experience. How the brain supports this process remains unclear. Here, we present a computational framework in which contextual

**Data availability statement:** All data and code used for running experiments, model fitting, and plotting is available on a public GitHub repository at https://github.com/JesPass/Contextual_RL, and has been deposited at 10.5281/zenodo.18401734.

**Funding:** AFM was funded by a UKRI Frontier Research Fellowship (an ERC Consolidator award) EP/Y034724/1 and a Medical Research Council (MRC) project grant number MR/W02005X/1 (www.ukri.org). J.P. was supported by the Wellcome Trust 4-year PhD in Neuroscience at UCL, grant number 222292/Z/20/Z (www.wellcome.org). The funders did not play any role in the study design, data collection and analysis, decision to publish, or preparation of the manuscript.

**Competing interests:** The authors have declared that no competing interests exist.

inference emerges from the interaction between two complementary processes. One process uses environmental features to infer context, a function often associated with the hippocampus, while the other uses recent behavioural outcomes to infer context, a function linked to prefrontal systems involved in monitoring actions and consequences. We show that combining these sources of information during learning allows agents to form stable, context-specific representations even when sensory cues are weak, noisy, or transient. Applying this framework to sequential decision-making tasks, we find that joint inference reproduces key behavioural signatures of contextual learning and generates context-dependent activity patterns resembling hippocampal splitter cells. Disrupting outcome-based inference during learning selectively impairs the formation of these representations. Together, our results suggest that interactions between feature-based and outcome-based inference may support flexible behaviour when contextual information is uncertain.

## Introduction

Humans and animals have a remarkable ability to learn and recall multiple optimal behaviours dependent on the current context [1–5]. While such contexts are often distinct environments that are easily distinguishable from one another, distinct contexts are also commonly found within the same environment where they must be differentiated by the presence of distinct features (such as smells, objects or sounds), or the distinct distribution of outcomes (such as rewards). This ability to identify distinct contexts and differentiate between them is crucial, as different behaviours are often needed to achieve the optimal outcome in each context. If there is imperfect information present at the time a decision is made within a context - referred to as partial observability - contextual inference is required to determine the current context. There are two major kinds of imperfect information: unreliable information and incomplete information [6–8]. Information can be unreliable due to being obscured by noise [9,10] or the presence of many other features [10–12]. In contrast, information can be incomplete at a decision-point, for example due to dependence on a spatially or temporally separated feature [13,14] or dependence on a sequence of observations [15,16]. Importantly, while the brain can effortlessly perform contextual inference, how this is achieved remains unclear. What is clear is that this requires complex interplay between multiple brain regions, in particular the prefrontal cortex and hippocampus [17,18]. Similarly, computational approaches are still limited in their ability to simultaneously learn to identify contexts and drive context-dependent behaviour [19,20].

Computationally, two main approaches have had success with learning to perform contextual inference. One approach is to use recurrence to maintain information related to the hidden context over time in the form of recurrent neural networks (RNNs) [10,13,21–23]. The other approach uses approximate Bayesian inference over model-based representations that learn explicitly how observations are linked to

each hidden context [1,24–27]. However, despite computational approaches being able to successfully maintain multiple context-dependent representations [22,26,28], during learning these approaches often struggle to identify the correct context upon which to learn, especially during more complex tasks. Specifically, models struggle to learn when the signal-to-noise ratio (SNR) of context-dependent to non-context-dependent observations decreases [10,13,23,29,30]. This inability to cope with low contextual SNR is a major limitation to applying these techniques successfully to complex environments as, given the noise and uncertainty of the world, often proportionately small changes signal a change in the current optimal behaviour. Recent evidence shows that this issue can be overcome by using externally provided context signals during learning, however how such supporting context signals can be generated and how they support learning remains unclear [10,31–34].

Interestingly, in the brain, regions that support contextual inference also seem to rely on support from other brain regions specifically during learning. In particular, prefrontal cortex (PFC) input to the hippocampus (HPC) is important during learning [15,35–37]. The HPC is thought to learn context-dependent, detail-oriented maps of features like sounds, location, time, and odours that allow for the prediction of upcoming features based on currently observed features [1,38–42], similar to RNNs [10,43] and Bayesian model-based representations [44,45]. In contrast, the PFC is often proposed to classify different contexts based on their distinct rules. More specifically, PFC is thought to differentiate between contexts based on the distinct outcomes that result from following each rule in each context [2,17,37,46–49]. As a result, within the brain it seems that outcome-based approaches in PFC may support the generation of a context signal that can support the learning of more complex, feature-based approaches in HPC.

Computationally, both feature- and outcome-based approaches can predict the current context based on their experience within the environment. An example of using features to infer the current context is adjusting the amount of flour you add to bread dough on the fly depending on the appearance and texture of the dough. However, if it is your first time making that specific bread and you do not know what the dough should look like, you can instead use the outcome of whether the resulting loaf is too dry or too wet to adjust the amount of flour you add next time. As is evident in this example, feature- and outcome-based approaches trade-off in their ability to predict contextual changes within a trial and in the complexity of the representations they use. Feature-based approaches learn complex, detailed representations of the environment, whereas outcome-based approaches focus on learning only about one parameter - outcomes associated with behaviour.

The greatly reduced complexity of learnt representations in outcome-based approaches compared to feature-based approaches allows outcome-based approaches to rapidly learn to distinguish different contexts [50–52]. However, by design outcome-based approaches can only react to changes in behavioural outcomes once they have been experienced. In contrast, although they are more complex, feature-based approaches are able to predict changes in outcome in advance of them being experienced, by identifying changes in the environment using the detailed representations they have created. On this basis, we hypothesised that during learning - especially where distinct contexts have increasingly overlapping features - the initial differentiation of contexts by feature-based models may benefit from the addition of a stable 'teacher' in the form of an outcome-based algorithm.

Therefore in this study we first characterise the performance of agents utilising each of these strategies in a simple, context-dependent task. We then go on to probe how these agents cope with tasks with decreasing contextual SNR, and ask to what extent combining both strategies mitigates these impairments. Finally, we test the generalisability of our findings by investigating how these algorithms perform on a series of common behavioural tasks used in the neuroscience community.

## Materials and methods

In order to investigate performance of feature- and outcome-based algorithms, as well as a joint model using both feature- and outcome-based information, on tasks requiring contextual inference, we used a cued T-maze, typically used across

both neuroscience and computational fields to model tasks with imperfect information at decision-points [13,14,33,53,54]. The cued T-maze consists of a central stem where a cue indicates which of two arms branching off of the central stem is rewarded (Fig 1). The cue is spatially separate from the primary decision-point at the end of the stem, resulting in this task requiring contextual inference to support decision-making as there is incomplete information present. Within this task, the features important for contextual inference are the cues, whereas the relevant behavioural outcomes are whether the trial is a left-turn rewarded trial type or a right-turn rewarded trial type.

To investigate how feature and outcome inference react to decreasing SNR of the contextual cue, we sampled two scenarios where the decreasing SNR of contextual information led to increasingly incomplete information at the primary decision-point. We either increased the distance between the cue and the primary choice-point [13,14] or made the context dependent on a sequence of multiple cues using tasks common to neuroscience called non-match to sample task and structural discrimination tasks [15,16]. We also tested the impact of increasing the unreliability of information to decrease the contextual SNR, by adding distractor cues common across both tasks around the contextual cue [10] (Fig 1).

## Behavioural task design

### Cued T-maze

We modelled the T-maze as consisting of discrete spatial locations in the form of one-hot vectors, on top of which we added cues as distinct features (Fig 2A). Each arm of the maze is made up of three distinct locations resulting in a state-space $S \in [s_1, s_2 \ldots s_9]$. Observations emitted at the start of the stem of the maze then take the form of $\phi(s_1) = [1, 0, 0, 0, 0, 0, 0, 0, 0, 0, 0]$. The predictive cues are overrepresented 4x in comparison to the locations and are

**Fig 1. Kinds of imperfect information used to investigate feature- and outcome-based algorithms.** Schematic of the cued T-maze used to investigate the impact of decreasing contextual SNR on algorithms. Schematics detail decreases in contextual SNR due to increasingly incomplete information due to either increasing spatial separation of the cue and the choice or dependence on a sequence of multiple cues, or due to increasingly unreliable information due to the addition of distractor cues around the contextual cue.

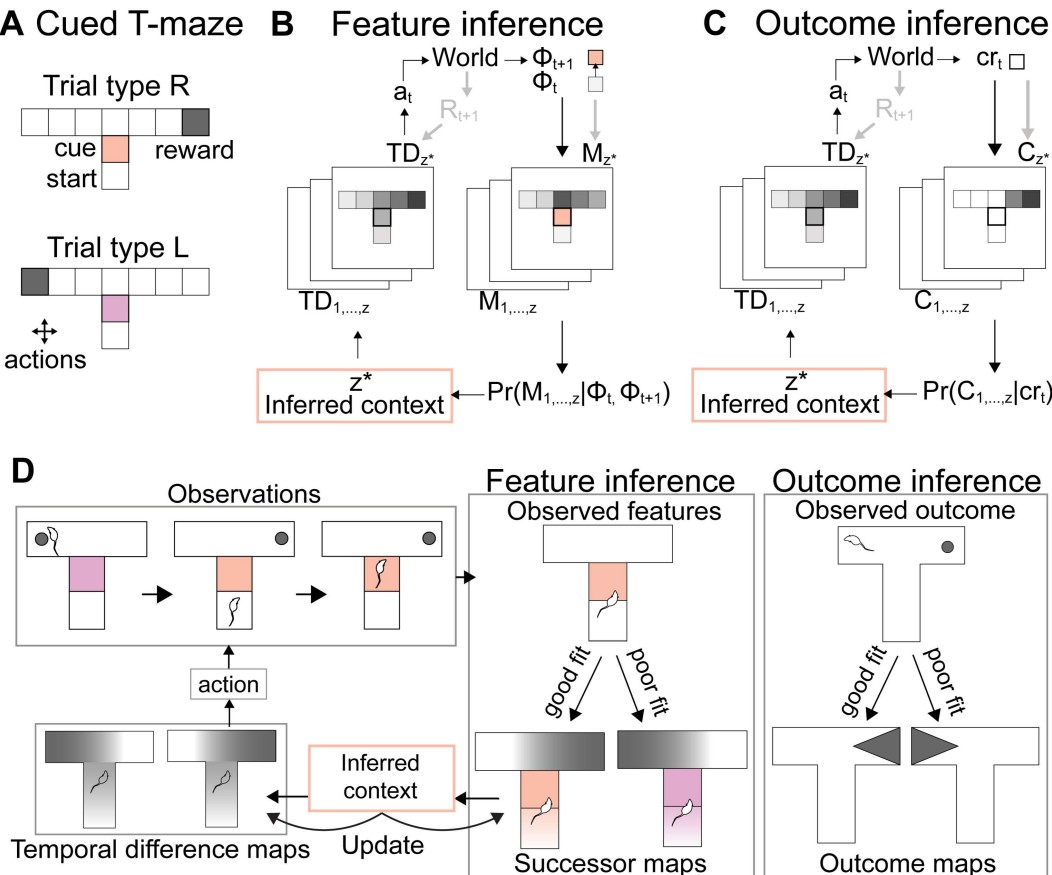

**Fig 2. Design of feature and outcome inference models. (A)** Setup of the cued T-maze task, showing different location and cue-based features, **(B)** Algorithm underlying feature inference showing how observed feature transitions $\phi_t$ to $\phi_{t+1}$ are compared with learnt successor feature maps $M$ using Bayesian inference to determine which context $z^*$ is currently most likely, followed by using the corresponding temporal difference map $TD$ to choose the current best action $a_t$, **(C)** Algorithm underlying outcome inference showing how observed outcomes $cr_t$ are compared with learnt convolved reward maps $C$ using Bayesian inference to determine which context is currently most likely, **(D)** Schematic of how the feature and outcome inference models react to a change in trial type, showing selection of the relevant map for feature inference following the cue and for outcome inference following the lack of reward, and how the inferred context allows for action selection using separate temporal difference maps.

present at the second location on the stem resulting in the observation $\phi(s_2) = [0, 1, 0, 0 \ldots, 4, 0]$ when in one context/trial type and $\phi(s_2) = [0, 1, 0, 0, \ldots, 4]$ when in the other context/trial type. The agent moves between states by selecting an action from the set of actions available to them $A \in [up, down, left, right]$. If the selected action involves running into a wall, the agent remains in its previous state. When the agent reaches the end of the correct arm, it receives a reward.

### Non-match to sample task and structural discrimination task

For the non-match to sample task and structural discrimination task we used a modified version of the T-maze design (Fig 1). Behaviourally these tasks are usually implemented as head-fixed tasks, where animals either must choose to lick from distinct left or right spouts [55,56] or preemptive licking on a set of rewarded trials and no preemptive licking on a set of non-rewarded trials is used as the read out for whether animals have understood the task [15,57–59]. Within our model the second case can be interpreted as rewarded trial types being equivalent to receiving reward at the end of the right arm of the maze and non-rewarded trial types being equivalent to receiving reward at the end of the left arm of the maze. As a

result actions taken that move the agent to the right can be thought of as preemptive licking, whereas actions taken that move the agent to the left can be thought of as not licking preemptively. Additionally, we only allowed the action $A \in [up]$ on the central stem of the maze to mimic the agent being presented with cues rather than physically moving between them. For the non-match to sample task, the predictive cues are overrepresented 2x and for the structural discrimination the predictive cues are overrepresented 4x.

### Training of algorithms

For the cued T-maze task, we trained agents on 20 block switches consisting of 50 trials each, such that the agents see each trial type 10 times. To investigate within-trial predictive inference, we then trained the agents on 500 random trials. For the joint inference algorithm, we used the joint prior and joint task estimate for the first 200 trials, which is equivalent to the first 4 block switches.

For the non-match to sample task, we pretrained agents alternating between blocks of the rewarded trial types until each trial type had been seen 6 times, with each block consisting of 50 trials. We then trained the agents on alternating blocks of all trial types until each trial type had been seen 5 times. To investigate within-trial predictive inference, we then trained the agents on 500 random trials. For the joint inference algorithm, we used the joint prior and joint task estimate for all of the block trials during training, but not during pretraining or random trials.

For the structural discrimination task, we pretrained agents alternating between blocks of the rewarded trial types until each trial type had been seen 3 times, with each block consisting of 50 trials. We then trained the agents on alternating blocks of all trial types until each trial type had been seen 3 times. To investigate within-trial predictive inference, we then trained the agents on 500 random trials. For the joint inference algorithm, we used the joint prior and joint task estimate for all of the block trials during training, but not during pretraining or random trials.

### Reinforcement learning problem statement

Reinforcement learning agents aim to maximise reward by interacting with an unknown environment. Here, they aim to solve the randomised cued T-maze, which represents a partially observable Markov-decision process (POMDP) defined by $D = (L, A, p, R, \Phi, \gamma)$ where $L$ is the hidden state space, $A$ is the action space and $\Phi$ is the observation space. For $l \in L$ and $a \in A$ the next hidden state $l'$ is given by $p(l'|l,a)$. The hidden state $l$ is defined by the history of past observations for $\phi \in \Phi$. The reward is defined as $R(l,a,l')$ and $\gamma \in [0, 1]$ is the discount factor which reduces reward values with distance into the future. The aim of the agent is to find a policy $\pi$, a set of mappings from hidden states to actions, that maximises expected discounted total future reward, known as value $V$, based on reward $R(l)$:

$$V(l) = \mathbb{E}_\pi \left[ \sum_{t=0}^{\infty} \gamma^t R(l_t)|l_0 = l \right]$$

The cued T-maze represents a specific instance of this problem where hidden states factorise into partially observable contexts $Z$ and fully observable states $S$ which maintain the same transition dynamics $p(s'|s,a)$ for all $s \in S$. The problem then reduces to a set of Markov-decision process (MDPs) $G^z = (S, A, p, R, \gamma)$ for $z \in Z$, where each context is defined by a specific instantiation of the reward function $r(s, a, s') \in R(s, a, s')$.

The optimal way to solve this problem is to infer $z$ to predict the current relevant behaviour. However, if $z$ cannot be predictively inferred based off of $\phi$ the problem reduces to a transfer learning problem as defined in [50] and [51] where the optimal behaviour is reactive to changes in $R(s,a,s')$ [52]. To model the behaviour associated with these two distinct strategies, we use an existing hippocampal model that attempts to predict the context within-trial and associated best behaviour, and an existing prefrontal cortex model that is reactive to behavioural outcomes.

## Feature inference model

To estimate value in this set of MDPs, first, the current relevant context must be identified. To infer $z$ we implemented an existing model of the hippocampus from [24] (Figs 2B, 2D, and S1A Fig). This model estimates its belief about the current context in the form of a belief state $b$, which is a distribution of probabilities over all possible contexts $b(z) = P(z|\phi)$. To do so, this model learns a set of trial-type specific successor feature maps $M^z$ where $M \in M^z$ is a predictive map of how observed features in $\phi$ are related to each other. Concurrently to learning these maps the model uses Bayesian inference over $M^z$ to determine $b(z)$ based on differing features between contexts.

Each successor feature map $M$ learns the predicted future occupancy of all other features based on the current column vector of observed features $\phi$. $M$ is an $i \times j$ matrix that linearly transforms the current feature observations to predict the future occupancy of all features given the current features, called successor features $\psi(s) = M\phi(s)$ for $\phi \in \Phi$ and $\psi \in \Psi$. We use subscript $i$ to denote the rows and $j$ to denote the columns, such that $M_{i,j} = M[i,j]$. Intuitively, $M_{i,j}$ is the predicted future occupancy of feature $i$ given feature $j$. $M$ is learnt using a simple delta rule $M_{t+1} = M_t + \alpha\delta g$ after every observation for $t \in T$ with the prediction error:

$$\delta = \phi(s_t) + \gamma\psi(s_{t+1}) - \psi(s_t)$$

Here, $g_i = \phi_{i,t}/x_t$ normalises the update depending on feature overrepresentation and the number of features observed $x$ such that entries in $M_{:,j}$ represent predicted future occupancies for $\phi_j = 1$.

From a probabilistic perspective the values $M_{i,j}$ are point estimates of the future predicted occupancies. We assume their distribution is Gaussian with variance $\sigma_j^2$, giving a multivariate normal of the form $M_{i,j} \propto N(M_{i,j}, \sigma_j^2)$. This allows us to use a flexible learning rate known as the Kalman gain $\alpha = k_t$ to anneal the learning rate as confidence in $M$ increases. The Kalman gain is calculated using a Kalman filter that tracks the uncertainty in model estimates $\Sigma_{t+1} = \Sigma_t - k_t h_t \Sigma_t$ versus the uncertainty in observations defined by the observation noise $\sigma_n^2$:

$$k_t = \frac{\Sigma_t h_t}{h_t \Sigma_t h_t + \sigma_n^2}$$

where $h_t = \gamma\phi_{t+1} - \phi_t$ is the discounted feature derivative. The Kalman gain decreases as the confidence in feature predictions increases, which is proportional to how often they have been observed.

To find the context map $z$ with the best fit to the observed features, Bayesian inference is used: $p(z|\phi_t) \propto p(\phi_t|z)p(z,t)$. The map that can best predict $\phi$ is the map where the future predicted occupancies of the observed state transition are best aligned, and thereby is the map that has the smallest prediction error. If $\delta = 0$, $\phi_t = \psi(s_t) - \gamma\psi(s_{t+1})$, so the probability of observing $\phi$ for each context map is estimated with:

$$p(\phi_t|z) = p(\phi_t|\psi(s_t) - \gamma\psi(s_{t+1}))$$

For inference in the feature-based algorithm we assume the variance is a constant predefined hyperparameter $\sigma_j^2 = \sigma_r^2$ to limit the impact of exploratory actions on context inference, as it allows for an improved tolerance for deviations from the current context map after learning.

For the prior a sticky Chinese restaurant process (sCRP) prior is used. Based on $N_z$, the number of previous observations assigned to each context $z$ over the last $y$ timesteps, a new context identity is generated according to the sCRP. The sCRP gives the probability of being in any of the previously experienced contexts or being in a new context:

$$p(z_t|z_{t-y-1:t-1}) = \begin{cases} \frac{N_z + \beta\delta[z_{t-1},z]}{\alpha+\beta+y}, & \text{if } z \text{ is previously sampled} \\ \frac{\alpha}{\alpha+\beta+y}, & \text{otherwise} \end{cases}$$

with the concentration $\alpha$, the prior belief as to how likely context switching is, and stickiness $\beta$, the prior belief as to how likely the context is to stay the same as on the previous step. The sCRP is based on the belief that although there is an infinite number of possible contexts, the agent is most likely to be in the context it has seen the most in the past weighted by recency. Since the prior is intractable, a particle filtering method from [51] is used for its estimation.

The inferred context is thus based on how well the features observed fit the features predicted by the learnt feature map for each context and is given by $z^* = \text{argmax}\, p(z|\phi_t)$.

## Outcome inference model

As described above, if $M^z$ has not been learnt well or cannot be used to infer the current context the problem nears a transfer learning problem, as inference based on $\phi$ is poor. We used an existing model of PFC from [51] that was designed to solve transfer learning problems to simulate this strategy (Figs 2C, 2D, and S1B Fig). The PFC model uses the same framework as the HPC model described above: it learns distinct maps of different contexts and uses Bayesian inference over them to determine the current context, making use of Kalman filtering to have a flexible learning rate and estimate the variance during Bayesian inference. The difference is in what information the maps contain. Instead of learning about observed features of the environment, the PFC model utilises a map of diffuse state-values to separate contexts where rewards are in sufficiently different states, meaning that different policies are necessary to reach rewards. The PFC model estimates $b(z)=P(z|cr)$ using Bayesian inference over a set of convolved reward value maps $C \in C^z$ that learn a local value estimate $cr$ for each $s$ based on reward locations. The PFC model learns about this environment as a tabular environment, where each state can be represented by a one-hot vector defined by current location. For example, observations emitted at the start of the stem of the maze take the form of [1,0,0,0,0,0,0,0,0] and the second location take the form of [0,1,0,0,0,0,0,0,0]. Briefly, outcome-based agents use a kernel of the form $F_\gamma = [\gamma^{-len}, \ldots, \gamma^{len}]$ with a filter length $len$ either side of the current state to calculate the convolved reward estimate $cr_t = F_\gamma r_{t-2len+1:t}$. Due to the filter, there is a temporal delay of $len$ states preceding inference. The agent updates its predictions for the currently inferred context map using a simple delta rule $C(s)_{t+1} = C(s)_t + \alpha(cr_t - C(s)_t)$, where $\alpha = \alpha_o k(s)$ and $k$ is calculated as above replacing $h_t$ with the current state $s_t$. The estimates the model learns again assume a Gaussian of the form $C(s) \propto N(C(s), \sigma^2(s))$. The same Bayesian inference mechanism is used as in feature inference and $p(z|cr_t)$ is determined with $p(cr_t|z) = p(cr_t|C(s))$. As a result, outcome maps have diffuse state value estimates allowing for clustering of different contexts based on different optimal policies. Of note, the outcome inference model does not receive cue features directly. This is in order to isolate outcome-driven contextual labelling from feature-driven predictive mapping, allowing their interaction to be analysed explicitly.

## Choice via temporal difference learning

We decided to separate the choice component from the inference component, to compare the ability of feature and outcome inference algorithms specifically in inference. For estimating state-action values and making choices we learn a set of temporal difference maps $TD^z$. Once a context identity has been selected a corresponding temporal difference model is used to pick the optimal action. The temporal difference $TD$ value for each state gives its discounted expected future reward and is updated with $\delta_t = r_{t+1} - TD_t\phi_t + \gamma TD_t\phi_{t+1}$ and the Kalman gain. The Kalman filter is used as above replacing $h_t$ with the current state features $\phi_t$. Future predicted reward is given by $Q(a|\phi_t) = TD(\phi_{t+1})$ where $\phi_{t+1}$ is determined using a one-step lookahead under the assumption that the agent has access to the transition dynamics of the environment $p(\phi_{t+1}|\phi_t, a)$. The action taken is chosen at random $\epsilon$ proportion of the time to allow for continuous exploration and with $a^* = \text{argmax}\, Q(a|\phi_t)$ the remainder of the time.

### Joint inference

During learning, the joint inference algorithm determines the current most likely task based on how well the observations $o_t$ fit both the successor feature maps and convolved reward value maps learnt for each context. It does so by averaging the contextual posterior from both feature and outcome inference:

$$p(z|o_t) = \frac{p(z|\phi_t) + p(z|cr_t)}{2}$$

Additionally, a joint prior is used where at each step the prior evolves once according to the feature dynamics and once according to the outcome dynamics. In contrast to maintaining a separate prior for each algorithm, a joint prior aids in ensuring that task estimates converge onto the equivalent maps within each algorithm and allows balancing of the different optimal hyperparameters across different algorithms. The joint inferred context is then used for selecting the corresponding context-specific TD map for action selection and updating corresponding feature and outcome maps (S1C Fig). After learning, the agent switches to using only feature inference.

### Other algorithms used for comparison

Here we provide a brief description of the models used for comparison.

### SR and SR1

The SR algorithms learn one successor representation map $M$, in the same way as the HPC algorithm does. In contrast to above, we do not learn a separate $TD$ map, but instead learn a reward vector for the SR to make choices. The SR allows decomposition of reward into the reward associated with each feature $w$ and the state features $R(s) = \phi(s)w$. The value function decomposes into future predicted occupancy of all features given the current features and the reward associated with each feature $V(s) = M\phi(s)w$. $w$ is learnt using a simple delta rule with learning rate $\alpha_r$. For one-shot SR $\alpha_r$ is set to 1.

### TD

The TD algorithm learns one temporal difference map $TD$ as described above.

### Other ways to combine algorithms explored

### Replay

During each attempt on a trial, the agent keeps track of the feature-transitions it observes. If the agent receives a reward at the end of the attempt, the trial type identity inferred by the outcome-based model is accepted as ground truth and the feature-transitions are replayed within said trial type in the feature-based model.

### Ideal observer

In the ideal observer model the outcome inference algorithm is told the actual trial type identity, which is assigned a probability of 95 percent. The identity of the other trial type is assigned a probability of 5 percent.

### Exploration

Each agent takes 10,000 steps via random walk in the environment without the predictive cues and rewards. The learnt SR and associated covariance make up the random-policy map.

## Parameters and hyperparameters

Outcome inference and feature inference model parameters were optimised using the initial task with the short maze stem, starting from the original parameters in [51] with an initial manual search for a reasonable range of values followed by a grid search within the found range. Parameters (Tables 1–3) remained the same for all other experiments, apart from the structural discrimination task. As talked about in the discussion, parameters for structural discrimination task (Tables 4 and 5) were readjusted following the same procedure as above, as there were 6 total contexts instead of 4.

## Statistics

Values plotted are the mean $\pm$95% confidence interval, unless there are outliers that skew the mean in which case the median is plotted (only in Fig 6A: cue-choice distance, (S3, S4, S5, S6, and S7B Figs): block switch attempts, (S11, S12, S13, S14C and S14E, and S15 Figs) blocks/random: number of attempts). Time-series data across algorithms is analysed using mixed ANOVA and bargraph data is analysed using ANOVA. For post-hoc testing, for ANOVAs we used pairwise Tukey tests and for mixed ANOVAs we used multiple T-tests with the Benjamini/Hochberg FDR correction. For comparisons across multiple factors, we used linear mixed models followed by post-hoc estimation of marginal means for across factor comparisons. * indicates $p < 0.05$. Statistical results are detailed in S1 Table.

**Table 1. Parameters across all models.**

| Parameter | Value | Description |
|---|---|---|
| $\epsilon$ | 0.2 | Explore-exploit ratio |
| $\gamma$ | 0.9 | Discount factor |
| $\sigma_n^2$ | 1 | Outcome noise variance |
| $p(z_0)$ | [1, 0.4, 0.1, 0, 0, 0, 0, 0, 0, 0] | Initial context probabilities |
| $P_0$ | *multinomial*$(p(z_0))$ | Initial particles |
| $n$ | 100 | Number of particles |
| $y$ | 10 | Particle context history |
| $z_{max}$ | 10 | Maximum task number |
| $\alpha_r$ | 0.7 | Reward learning rate |

**Table 2. Parameters for feature inference.**

| Parameter | Value | Description |
|---|---|---|
| $\alpha$ | 0.5 | Concentration |
| $\beta$ | 5 | Stickiness |
| $\sigma_0^2$ | 1.6 | Initial residual variance |
| $\sigma_r^2$ | 1.6 | Residual variance for inference |

**Table 3. Parameters for outcome inference.**

| Parameter | Value | Description |
|---|---|---|
| $\alpha$ | 0.1 | Concentration |
| $\beta$ | 0.7 | Stickiness |
| $\sigma_0^2$ | 1.6 | Initial residual variance |
| $\alpha_0$ | $\frac{2}{3}$ | Learning rate weighting |
| *len* | 2 | Filter length |

**Table 4. Parameters for feature inference in biconditional discrimination.**

| Parameter | Value | Description |
|---|---|---|
| $\alpha$ | 0.5 | Concentration |
| $\beta$ | 8 | Stickiness |
| $\sigma_0^2$ | 1.6 | Initial residual variance |
| $\sigma_r^2$ | 1.6 | Residual variance for inference |

**Table 5. Parameters for outcome inference in biconditional discrimination.**

| Parameter | Value | Description |
|---|---|---|
| $\alpha$ | 0.8 | Concentration |
| $\beta$ | 0.7 | Stickiness |
| $\sigma_0^2$ | 1.6 | Initial residual variance |
| $\alpha_o$ | $\frac{2}{3}$ | Learning rate weighting |
| $len$ | 2 | Filter length |

## Compute power

The compute power needed to generate the data in this paper from these algorithms is around 300 hours using 200 cores each with 5GB RAM.

## Code and data availability

Model code, analysis code and data generated by the models is available at https://github.com/JesPass/Contextual_RL.

## Results

Our main goals were to investigate i) the mechanism underlying deficits in learning to perform contextual inference that occur as contextual SNR decreases and ii) whether such deficits could be alleviated by incorporating outcome-based inference during learning. To investigate the performance of outcome- and feature-based approaches, we trained and probed the learning of these algorithms on a simple cued T-maze paradigm often used in both neuroscience and computational experiments investigating contextual inference (Fig 2A). In this task a distinct cue at the beginning of a T-shaped maze indicates whether the left or right arm is rewarded [13,14,33,53,54]. Additionally, this task has been shown to engage hippocampal-dependent processing, particularly in variants where the cue is not available at the choice point, such that correct behaviour requires maintaining or inferring contextual information beyond the cue presentation [14,60]. As a result, this choice of task facilitated comparison of the performance and workings of our algorithms with neuroscience data (see also Fig 8 for comparison with other common neuroscience tasks).

We modelled the T-maze task by representing each arm of the maze as 3 discrete locations - leading to 9 total discrete locations. We then represented the cue as an additional discrete feature at one single location in the starting arm (Fig 2A). For example, the first location in the map was represented as [1,0,0,0,0,0,0,0,0,0,0,0] and the second location with the cue was represented as $[0, 1, 0, 0 \ldots, 4, 0]$ when in one trial type and $[0, 1, 0, 0, \ldots, 0, 4]$ when in the other trial type. The factor of 4 scales the cue feature relative to spatial state features to ensure sufficient discriminability between trial types in the baseline regime. To perform the task, an agent must choose actions to transition between discrete locations. After every step, the agent observes its current location and whether there is a cue present. Each 'attempt' ends when the agent reaches the end of the left or right arm, after which the agent is teleported back to the starting state. The agent has an unlimited number of attempts to make the correct choice and once the correct choice is made it moves into the next trial.

The cued T-maze specifically requires contextual inference as the cue is not presented at the choice point, resulting in the state at the choice point being a hidden state.

We focused on a Bayesian model-based approach, to allow us to easily probe how learnt representations support behaviour. Specifically, to model feature-based strategies, we used a Bayesian successor feature algorithm, which we refer to as 'feature inference' (FI). This model infers and predicts optimal behaviour within-trial based on differing features of the environment using context-specific successor representations (SRs) [24]. To model outcome-based strategies, and to investigate whether these can be used to support learning of feature inference, we used a similar model-based Bayesian approach, which we refer to as 'outcome inference' (OI). In contrast to feature inference, this model learns context-specific diffuse state-outcome maps to identify contexts [51]. As outcome inference focuses its attention solely on state-outcomes, which define the optimal behaviour, we hypothesised that it would be well suited to support initial learning of context-specific SRs for within-trial predictive inference.

To solve this task, both FI and OI learn a set of context-dependent maps of the world, which they use to identify the current context. FI learns a set of SRs $M$ and OI learns a set of convolved reward maps $C$. The SRs learn the predicted future occupancy of all features of the environment given the currently observed features (see methods) (Fig 2B). In contrast, the convolved reward maps $C$ learn an average estimate of reward in each location using a filter that weights any rewards located either side of the current location (see methods) (Fig 2C). As a result, FI receives the full feature vector including the location and cue as the observation input, whereas OI receives only the location component. For example, for OI observations emitted at the start of the stem of the maze take the form of [1,0,0,0,0,0,0,0,0,0] and the second location take the form of [0,1,0,0,0,0,0,0,0,0]. Importantly, at the same time as they are learning these maps, both algorithms also use Bayesian inference to compare the current observations from the environment with their current maps of the world to infer which context $z$ they are most likely in. Specifically, FI compares the features $\phi$ observed by the agent prior to and following an action and compares how well the observed transition fits with the predictions encoded in each successor map $M$. FI does so by estimating the probability of observing $\phi$ for each context map with:

$$p(\phi_t|z) = p(\phi_t|\psi(s_t) - \gamma\psi(s_{t+1}))$$

where $\psi(s_t)$ represents the predicted future occupancy of all features given the current features, called successor features. In contrast, OI compares the observed convolved reward $cr$ with the predicted $cr$ from each convolved reward map $C$ by estimating the probability of observing $cr$ for each context map with:

$$p(cr_t|z) = p(cr_t|C(s))$$

Based on their current observations, the agents can either infer whether the current observations align well with an existing map; or if not, the agent can create a new map.

FI or OI cannot on their own be used to select actions, as FI does not learn about rewards and OI learns only a local representation of reward. As a result, in addition to this overall architecture, for each context a corresponding temporal difference $TD$ map is learnt for action selection within each context (Fig 2B and 2C). For FI specifically, an alternative would be to learn a reward vector indicating which features are rewarded to allow for predicting the best current action. Here, we used $TD$ maps for both strategies to (1) allow both FI and OI to use the same mechanisms for decision making to facilitate comparison of the two inference strategies and (2) isolate contextual inference from action selection. The $TD$ maps learn value gradients across the locations and features of the environment using the prediction error $\delta_t = r_{t+1} - TD_t\phi_t + \gamma TD_t\phi_{t+1}$, thereby allowing the agent to evaluate which action brings them closer to the reward (see methods for more detail). Once a context has been inferred, the context identity is passed to the corresponding context specific $TD$ map to select the next action. The predicted value of each action is given by $Q(a|\phi_t) = TD(\phi_{t+1})$ where $\phi_{t+1}$ is determined using a one-step lookahead under the assumption that the agent has access to the transition dynamics of the environment $p(\phi_{t+1}|\phi_t, a)$. The

action taken is chosen at random $\epsilon$ proportion of the time to allow for continuous exploration and with $a^* = \text{argmax } Q(a|\phi_t)$ the remainder of the time. Finally, the corresponding *TD* and *M* or *TD* and *C* maps are updated.

Importantly, the agents are not provided with any representations of the environment and do not receive any external context signals, aside from the environmental observations that indirectly indicate the hidden context. As a result, the main challenge for the agents is that they must simultaneously learn context-dependent representations, and use these representations to infer the current context and guide behaviour. As a simple comparative metric, if the algorithms learn accurate representations of both contexts, within the cued T-maze FI should be able to predict the current context within-trial based on the cue, whereas OI should be able to react to the absence of reward on the previously rewarded arm to infer the current context (Fig 2D).

### Feature and outcome inference outperform alternative strategies during cued switches in a T-maze

We first asked whether FI and OI can solve contextual inference problems under relatively high contextual discriminability. As a baseline, we examined whether the agents could solve a simple cued T-maze task that has previously been used in studies of contextual inference, before systematically increasing task demands by manipulating contextual SNR [13,29,30]. The simple cued T-maze configuration provides a controlled regime in which contextual evidence is strong and spatial overlap between contexts is minimal. This baseline allows us to then investigate how each model type is influenced by the amount of overlap between environments in different contexts. We will do so either by i) adding distractor cues around the predictive cue or ii) increasing the distance between the predictive cue and the choice.

In order to investigate whether agents and animals can perform within-trial predictive, cue-based inference in the cued T-maze we present trial types from each context in a random order, as in the computational and rodent literature [13,14,29,30,54,60]. However, in laboratory settings an initial learning phase is required for animals to learn this task [14,54,60,61]. Pre-training for this task often involves a combination of pre-training in the form of shortening the distance between the cue and the choice and blocking off the incorrect arm [14,54,60,61]. However, as we wanted to investigate how learning could occur when presented with the full task, we used an alternative method used to train animals on the cued T-maze consisting of using a blocked learning phase, where trials are arranged in blocks of the same trial type [10]. This is a common training technique used to teach similar contextual inference tasks in both the rodent and human literature [9,46,62–64], as well as in the computational literature, specifically for training Bayesian SR algorithms [1,24]. Guided by this literature, we first trained FI and OI agents to perform the cued T-maze in a blocked learning phase, where trials were arranged as blocks of the same trial type, before moving agents on to the full task with randomised trials.

Within each trial, agents were allowed as many attempts as necessary to reach the rewarded location, often referred to as 'correction trials'. An attempt was defined as an agent reaching the end location within either of the arms, after which, if the end location was incorrect, the agent was teleported back to the starting state on the central stem of the maze for the next attempt, or, if the end location was correct, the next trial. Once agents had completed 50 trials in one context, the context was switched such that the cue was different and the reward was located on the opposite arm. We trained agents on this version of the task for 10 blocks (500 trials). Next, we allowed these agents to continue to perform the task, and investigated their performance on a further 10 blocks to see how quickly each algorithm reacted to changes in the trial type (Fig 3A and 3B). We found that both FI and OI models learnt to perform the block switches well, with OI requiring 2–3 attempts to reach the correct location following a trial type switch and FI requiring 1–2 attempts (Fig 3C and 3D). This rapid adjustment is qualitatively consistent with reversal behaviour observed in rodent navigation paradigms, in which adaptation to contingency changes typically occurs over several trials, depending on task structure and cue availability [9,10,46].

To test whether Bayesian inference is necessary within our modelling framework for solving the cued T-maze contextual inference problem, we next compared the performance of these models with the performance of their individual components. We specifically compared the algorithms to an SR algorithm and TD algorithm that both learnt only one map

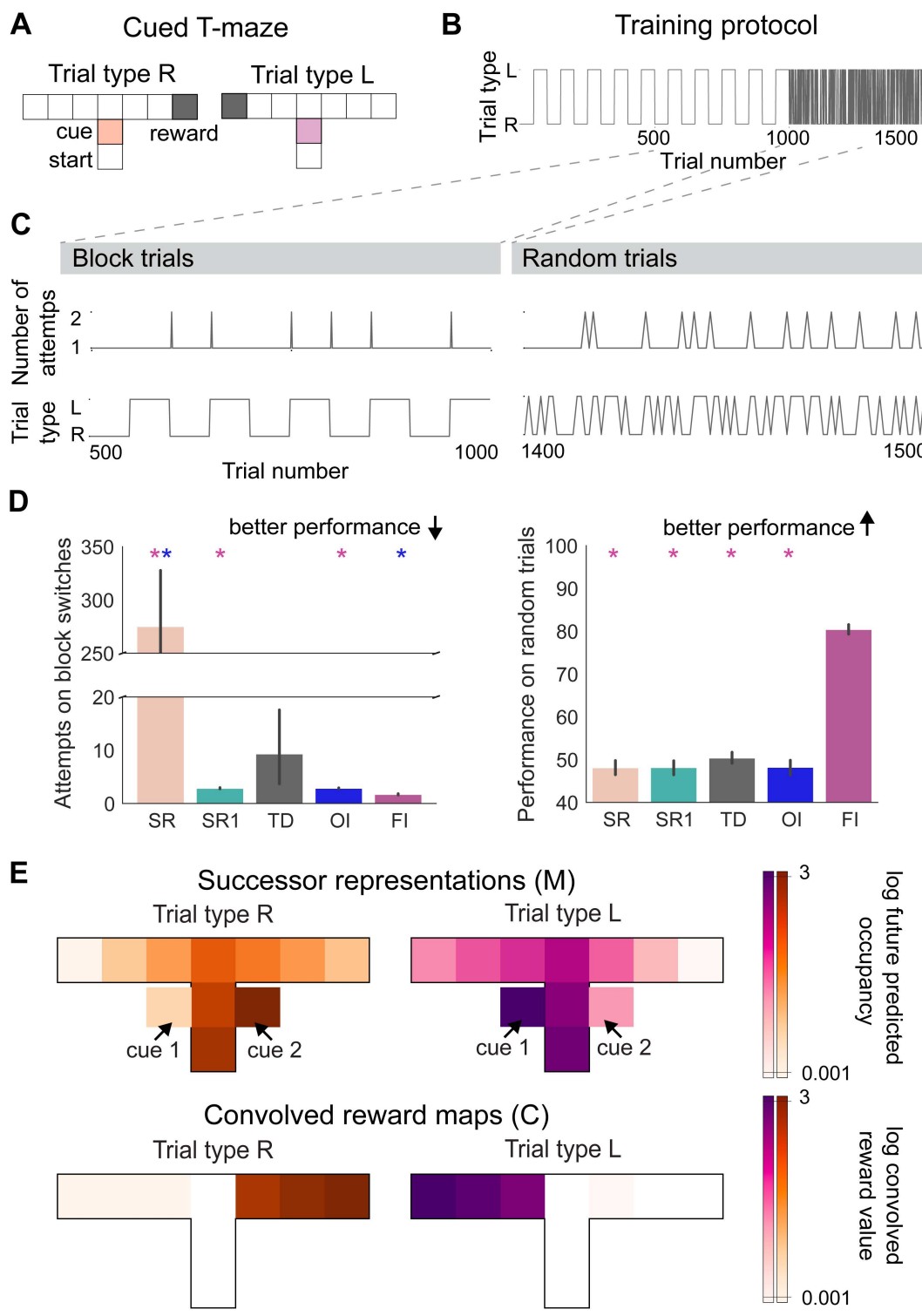

**Fig 3. Feature and outcome inference learn distinct strategies to solve a cued T-maze. (A)** Setup of the cued T-maze, showing distinct cues and reward locations for distinct trial types, **(B)** Training setup for the paradigm, showing trial identity for 1000 trials during block switches followed by 500 random trials, **(C)** Performance of an example feature inference agent on the last 10 block switches (left) and last 100 random trials (right) used to quantify their performance, **(D)** Performance of feature inference and outcome inference in comparison to other RL agents on block switches (left) and random trials (right), **(E)** Trial type specific SRs learnt by the feature inference algorithm showing distinct predicted future occupancy when the agent is in

the starting state, averaged over all agents on the last 100 random trials (log-scale), compared with trial type specific convolved reward maps learnt by the outcome inference algorithm, showing distinct expected behavioural outcomes, averaged over all agents on the last 500 trials during blocks of trials. Predicted future occupancy of locations is indicated within the T-maze and predicted future occupancy of cue 1 associated with trial type L is indicated to the left of the location it occurs in and cue 2 associated with trial type R is indicated to the right of the location it occurs in. * indicates p < 0.05, statistical results are detailed in S1 Table.

of the environment. Within the SR algorithm, we compared two different versions i) where the reward vector was either updated incrementally (SR) or ii) where the learning rate was set to 1 such that rewards were learnt and unlearnt within one experience (SR1). We found that both feature and outcome inference performed better than the algorithms that do not use Bayesian inference on block switches (Fig 3C,D). This increase in performance of inference agents is due to the SR and TD algorithm both needing to gradually unlearn the reward location and relearn the new reward location following a trial switch. Even in SR1, following the absence of reward the agent must re-explore the whole environment to find the new reward location, as it has no memory of previous reward locations.

### Within-trial predictive inference is required to solve a cued T-maze

We next wanted to test the extent to which each algorithm could support within-trial predictive inference. To do this, we exposed each agent that had been trained on the block switch task described above, to a version of the task that required predictive cue-based inference within each trial, where the context (and hence appropriate behaviour) on each trial was chosen at random. Therefore in this instance, past outcome is not informative for behaviour, and so we reasoned that OI agents would not be able to perform this version of the task. We let each agent perform an additional 400 trials before investigating their performance over 100 further random trials (Fig 3B).

Consistent with this reasoning, FI agents could accurately predict the reward location based on the cue within-trial around 80 percent of the time, while OI agents could not. This level of performance lies within the range typically observed in rodent cue-guided navigation paradigms, where accuracy often stabilises below ceiling depending on task structure and contextual ambiguity [10,14,54]. Indeed when we compared performance to all other model types (SR, SR1, TD), only FI algorithms were able to perform within-trial predictive inference (Fig 3C and 3D).

We next investigated what specific representation allowed FI and OI agents to perform the task. In this architecture, contextual inference gates the updating of individual successor representations, enabling the learning of separate successor representations for each context, including distinct representations of cue identity and associated behaviour. This can be seen by comparing the average predicted future occupancy of each feature when the agent is in the starting location across each context (Fig 3E). Similarly we found that OI agents formed distinct convolved reward maps for each context, learning distinct representations of the expected reward along the arms of the maze leading to the reward location (Fig 3E).

Together these initial results confirm previous reports [51] that OI can learn to use an outcome-based strategy that is effective at solving tasks that contain blocks of repeating trials, but cannot solve random trial presentations, where the current best behaviour is independent of past outcomes. In contrast, FI can learn to use a within-trial predictive strategy, offering an improved performance on blocks of trials, as well as the ability to solve random trials.

### Performance of feature inference degrades as contexts become more similar

Our results so far suggest that in principle, FI is well-suited to perform within-trial predictive inference in the cued T-maze task. We next looked at the effect of decreasing the contextual SNR, by increasing environmental overlap between contexts. We tested this with two complementary manipulations to the task structure. First, we created a series of different mazes, where we incrementally added an increasing number of 'distractor' features that are present in both contexts around the cue (Fig 4A) [10,65–67]. In other words, while in the base task the cues may have been A vs B, a version with

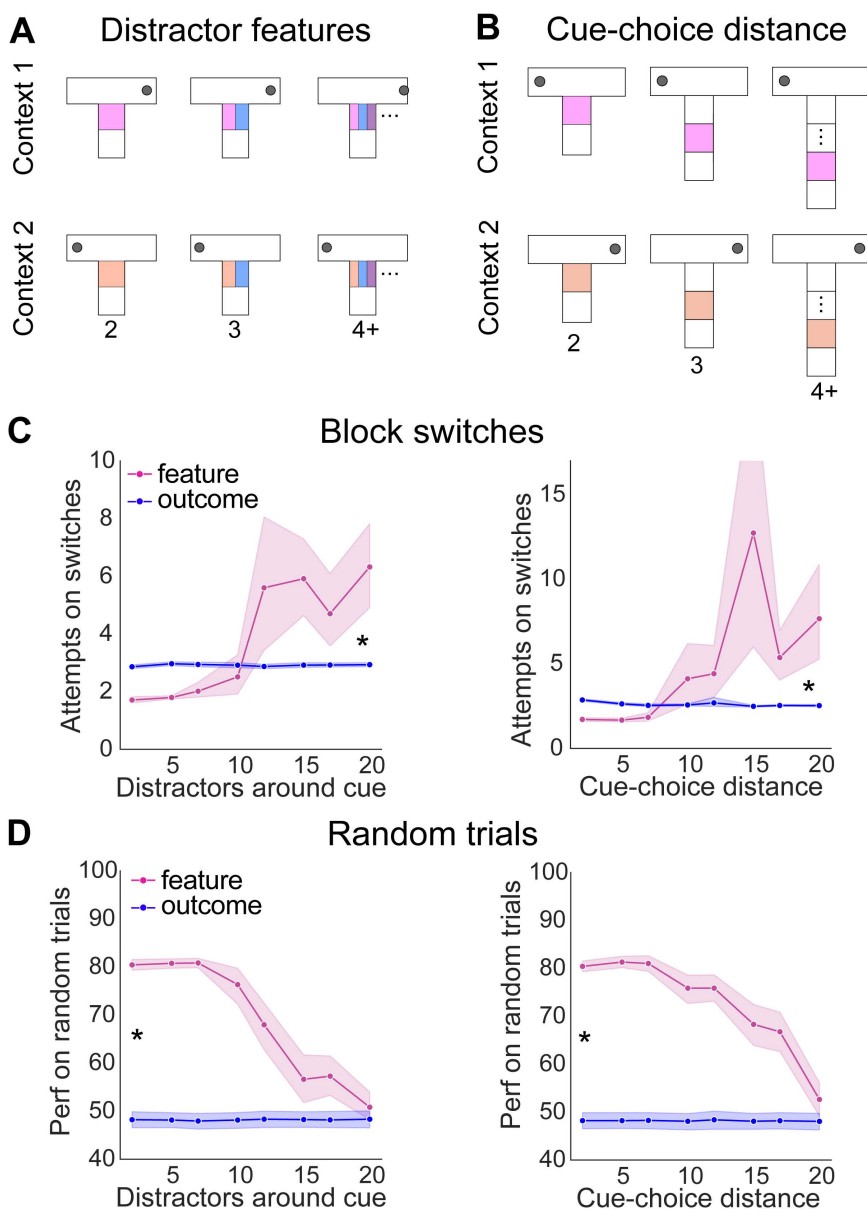

**Fig 4. Increasing overlap between contexts reduces performance of feature inference, but not outcome inference. (A)** Decreasing the contextual SNR by adding multiple overlapping features around the cue in each context or **(B)** extending the distance between the cue and the choice-point, **(C)** Performance of feature and outcome inference as contextual overlap increases on block switches and **(D)** random trials. * indicates p < 0.05, statistical results are detailed in S1 Table.

3 distractor cues has 4 cues in each maze in total, three of which overlap: XYZA vs XYZB. To complement this, we also created a series of mazes where instead we incrementally increased the distance between the cue and the choice-point, without changing the number or identity of the cues (Fig 4B) [10,14,53,54]. In this task the increased overlap between the two contexts arises from the increasing number of locations in the centre arm, that due to their lack of distinguishing features are highly similar across contexts.

Using this approach, we found distinct effects of increasing environmental overlap across FI and OI agents. These differences were independent of the means by which the overlap increased. During the block switching stage of the task, the number of attempts required by FI agents to reverse increased consistently with increasing environmental overlap between contexts (Fig 4C). In contrast, we found that the number of attempts required for reversal on block trials under OI was extremely consistent, independent of the amount of environmental overlap between contexts (Fig 4C).

Similar to the block switching stage, the performance of FI on random trials decreased steadily to chance with increasing contextual overlap (Fig 4D). This was in contrast to OI agents remaining unable to perform above chance independent of the amount of contextual overlap. Together, these results indicate that during learning, OI is robust to decreases in contextual SNR, resulting from increasing overlapping features. This suggests that OI may be well-suited to provide a context signal to FI agents that otherwise struggle during learning with low SNR.

**Supporting feature inference with outcome inference during learning rescues performance at low SNR**

Our data suggest that FI agents increasingly struggled to perform the T-maze task as the two contexts became more similar. In contrast, OI agents were able to stably learn to separate the two contexts, independent of their overlap. Therefore we reasoned that supporting FI with OI during learning may overcome the limitations of FI, by supporting more accurate initial learning.

To test this, we built a joint inference architecture where OI contributes to context assignment during early learning (Fig 5A). To implement this, we began by generating joint agents containing both FI and OI algorithms, and allowing the algorithms to interact with one another only during the initial learning phase - for the first 200 trials of the task. During these trials the agent created a 'joint estimate' of which context is most likely by averaging the posterior probabilities over contexts computed independently by FI and OI, as well as having a 'joint estimate' of the prior, where both FI and OI contribute to the evolution of a single prior (Fig 5B). We chose to average the probabilities and have a joint prior over the first 200 trials as a proof of principle experiment, as it allows us to investigate whether the simple metric of averaging implemented for very few trials at the start of learning is sufficient for improving performance. This jointly inferred context identity is then used for selecting the corresponding context-specific *TD* map for action selection and to update the corresponding outcome and feature inference maps. Importantly, the FI algorithm does not receive any direct input from the OI algorithm. After the first 200 trials, the agents then revert to a solely FI strategy, and no longer incorporate information from OI.

When we investigated this joint inference model, in contrast to FI alone, we found this algorithm had stable performance across all contextual SNRs, both in the case where SNR decreased with increases in distractor features and in the case where SNR decreased with increases in distance between the cue and the choice (Fig 5C and 5D). Crucially, although we only allowed the interaction of the two agents for from first 200 trials of the block switches, this stable performance lasted throughout the remaining block switches, and also resulted in stable performance during random trials. Thus, supporting FI during learning with OI rescues deficits in performance with low SNR, even after input from the OI algorithm is no longer present.

Importantly, the ability to incorporate OI into FI estimates can occur in multiple ways - not just via real-time access to both estimates to drive contextual inference. Therefore, we also tested whether this ability to support feature inference could be achieved using multiple alternative implementations of the interaction. In particular we focused on a model that uses replay of contexts inferred by outcome inference after each trial (S2A Fig), and a model that uses ideal observer estimates of the current context (S2B Fig). In both cases we found that such approaches did not consistently overcome the deficits of feature-based learning. Therefore, while multiple mechanisms could support this joint estimate, it seems that real-time access to both outcome and feature based contextual estimates is particularly beneficial for accurate contextual inference at low SNR.

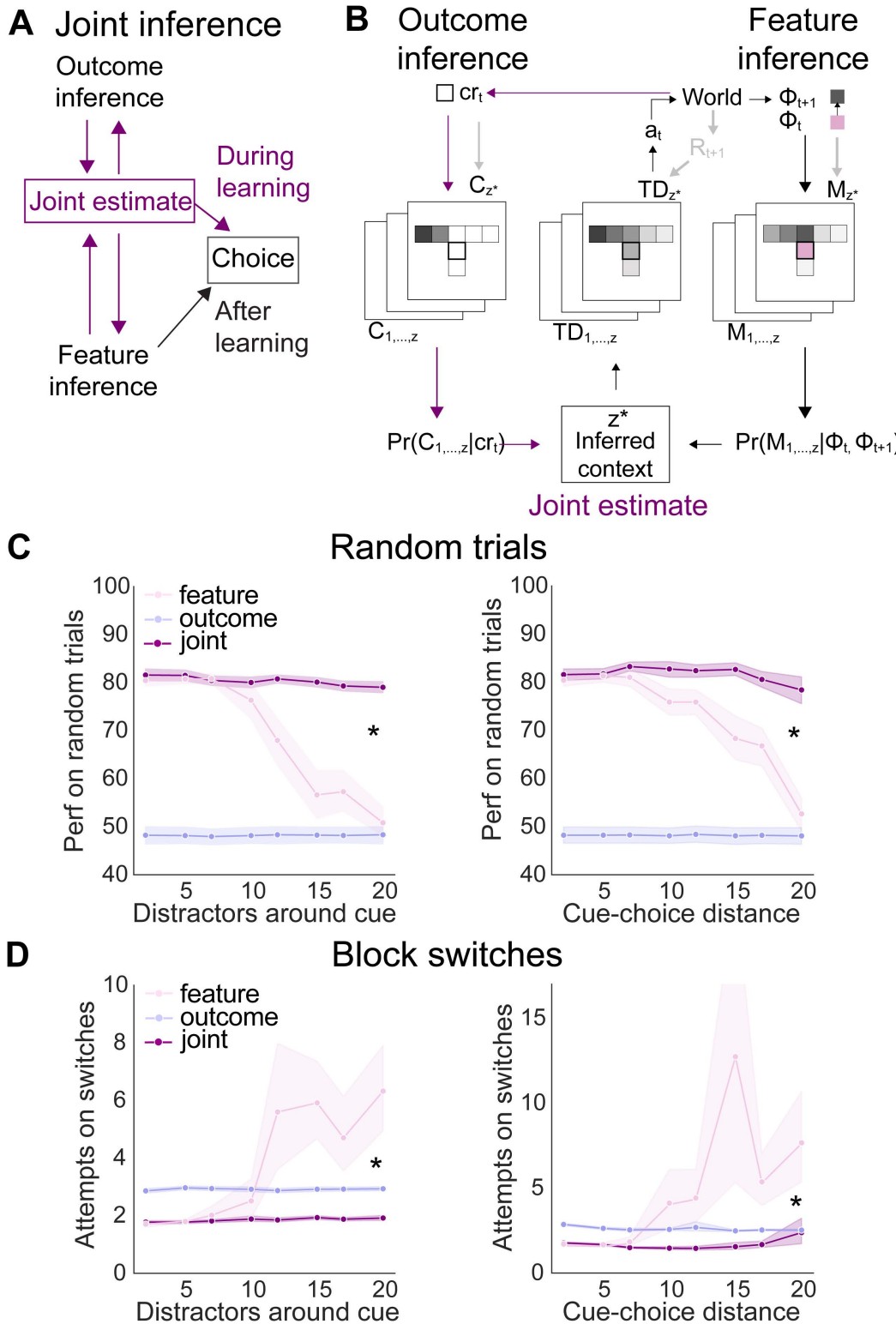

**Fig 5. Supporting feature inference with outcome inference during learning rescues performance with increasing contextual overlap. (A)** Schematic of joint algorithm showing that during learning outcome inference is used to generate a joint context estimate, **(B)** Algorithmic implementation showing how the joint estimate is determined during learning, **(C)** Performance of joint inference on random trials as contextual overlap increases

via distractor features around the cue or increasing cue-choice distance, **(D)** Performance of joint inference on block switches as contextual overlap increases via distractor features around the cue or increasing cue-choice distance. * indicates p < 0.05, statistical results are detailed in S1 Table.

## Access to outcome inference stabilises the learning of feature-based representations

Our data confirm that the presence of OI information during learning greatly facilitates the use of FI, even long after this OI information is no longer available. We next wanted to investigate *how* the addition of OI to the FI algorithm supported this facilitation of learning. We found that in the block learning phase, FI, but not OI or joint inference, struggled to maintain performance as environmental overlap between contexts increased. We reasoned that the distinction between algorithms is that in FI it is necessary to learn to predict the future occupancy of all of the features in the environment, whereas OI only learns to predict one feature - the reward. As overlap between contexts increases, the features of the environment become more similar between contexts, but the observed rewards do not. Under these conditions, FI agents are more likely early in learning to assign observations to the incorrect context, because feature similarity reduces contextual discriminability. This misassignment reduces the separation between context-specific predictive maps, which prevents the differentiation of contextual representations.

To understand how OI contributes to context assignment during early learning via joint prior and posterior estimation, we first looked at the learning curves of agents across blocks of trials and found that as the number of distractors or cue-choice distance increased the number of attempts FI required to switch its behaviour specifically on the first block change, where the agents experienced a switch in trial type for the first time, increased drastically (S3, S4, S5A, S5B, and S6A Figs). Contrastingly, the number of attempts for OI across the first and following block switches remained consistent independent of the contextual SNR. When supporting FI with OI in the joint algorithm, the number of attempts required on the first block switch remained consistent across different contextual SNR. This improvement in the number of attempts required remained even after OI was no longer present from trial 200 onwards (S3, S4, S5B and S6A Figs). This improvement in block switches from FI to the joint algorithm co-occurred with an improvement of stability of behaviour within blocks (S3, S4, S5A, S5C and S6B Figs). On the basis of these learning curves, we hypothesised that the positive effect of OI on FI in the joint model occurs due its stabilising influence on the first block switch.

To investigate whether the primary benefit of supporting FI with OI occurred due to stabilisation of behaviour on the first block switch, we tested allowing OI to interact with FI for different lengths of time during learning. We found that allowing OI to support FI for anywhere from the first 100 block trials, which includes the first 50 trials preceding the first block switch and the 50 trials after, to all block trials and the first 250 random trials, did not impair performance on the last 10 block switches or the last 100 random trials for the baseline task, the maximum number of distractors (S7A Fig) or the longest cue-choice distance (S7B Fig) tested above. Additionally, for the longest cue-choice distance it was even sufficient for OI to support FI only on the first 50 trials, not including the first block switch (S7A Fig). Together, these results indicate that OI supporting the first block switch allows for a stable representation of the context following the trial type switch to form, either by directly reducing the number of attempts required following the trial type switch or indirectly by supporting the formation of a stable contextual map within FI during the first 50 trials.

Interestingly, we also found that allowing OI to continue to interact with FI during the last 100 random trials as well prevented expression of the FI feature-based strategy on these random trials, leading performance to drop to chance (S7A and S7B Fig). Together, these results suggest that the interaction between OI on FI must be tightly regulated based on current task demands and that the primary benefit from OI supporting FI comes from its support of the first block switch.

To investigate further how OI allows for a cleaner transition between inferred contexts on the first trial type switch, we compared the evolution of the average posterior probabilities of the original context map based on OI and FI on the first attempt following the first block switch with estimates from OI and FI when they were allowed to interact in the joint algorithm. We found that within FI the cue led to a sharp drop in the inferred probability of the, now incorrect, original context

map, however the increasing similarity of the contexts with the addition of either more distractors around the cue (S8 Fig) or more distance between the cue and the choice (S9 Fig) prevented this drop from occurring. In contrast, OI maintained a high posterior probability throughout the trial with a sharp drop following the absence of expected reward, independent of the number of distractors (S8 Fig) or cue-choice distance (S9 Fig). As a result, in the joint inference algorithm the posterior probability is primarily driven by the joint prior, which results in the posterior of FI nearing OI, and the joint estimate made using both posteriors (S8 and S9 Figs). We found that allowing FI and OI to interact early during learning restored the large drop in the probability of the map from the previous, now incorrect, context following the cue on the first attempt following the last trial type switch (S8 and S9 Figs). Because the OI algorithm produces a sharp reduction in posterior probability following omission of expected reward, independent of contextual SNR, averaging its posterior with that of FI during early learning produces a clearer joint context estimate. This joint estimate reduces the number of attempts required on the first block switch.

### Outcome inference facilitates correct feature updating on the first switch

We reasoned that this OI support of FI may be uniquely important during this first trial switch beacuse when overlap between contexts is high, the FI algorithm incorrectly incorporates new observations in the original context map, rather than creating a new context map from these new observations. Consistent with this, we found that agents using FI displayed a large increase in the amount of incorrect updates to the original context representation as overlap increased (Fig 6A). In contrast, and consistent with our previous data, this incorrect updating did not occur when using OI or joint inference, and these differences between FI, OI and joint inference were independent of whether overlap was increased by increasing the number of distractor features, or increasing the distance from the cue to the choice point (Fig 6A).

We next investigated what influence this incorrect updating had on the representations of context. To do this, we plotted the predicted future occupancy of each environmental feature when the agents are at the start of the maze both before and after the first trial type switch (Fig 6B). We found that at the end of the first block, preceding the first trial type switch, there was a clear distinction between the expectation of cue 1 and cue 2 (Fig 6B). We then compared the effect of the first trial type switch on the representation of both cues in the FI and joint inference algorithms. Interestingly, when using FI, the incorrect cue became incorporated in the original context representation (note high expected cue 2, Fig 6B). However, this was not the case when FI was combined with OI in the joint algorithm (note that cue 1 continues to be the most expected, Fig 6B). Therefore consistent with our prediction, FI agents incorporate new observations into the wrong representation after the first switch in context.

To investigate whether this degradation of context-specific maps when using FI was long-lasting, we looked at the representation of the correct and incorrect cue within random trials. To do so, we looked within the FI maps for each task that had the highest average probability at the choice-point on correct trials in the last 100 random trials. We then looked at how well the transition from the starting location $s_t$ to the next location with the correct or incorrect cue $s_{t+1}$ was represented within each context map using $\psi(s_t) - \gamma\psi(s_{t+1})$, which tells us how well the correct/incorrect cue is predicted in the starting location within each context map (see methods for more detail). If the predicted future occupancy of the cue in the starting state is equal to the actual observed cue this value goes to 0, if the predicted future occupancy of the cue is larger than the actual observed cue this value is larger than 0, and if the predicted future occupancy of the cue is less than the actual observed cue this value is smaller than 0. Interestingly, we found that the degradation of context-specific maps when using FI was long-lasting: on random trials the FI agents had a decreased representation of the correct cue and an increased representation of the incorrect cue in both context representations as the amount of environmental overlap increased. This was despite these trials occurring many hundreds of trials after the first trial type switch. This long lasting degradation was also mitigated by use of a joint agent, despite again the access of this agent to OI information being removed many hundreds of trials earlier (Fig 6C).

**Fig 6. Supporting learning with outcome inference improves initial formation of context-dependent representations leading to long-lasting improvement in performance. (A)** Number of incorrect updates of the context map during the first trial type switch, **(B)** Evolution of the average context map's predicted future occupancy when the agent is in the starting state (log scale), at the end of the first block and the end of the second block of trials

for joint and feature inference agents, **(C)** Underrepresentation of the correct cue (left) and overrepresentation of the incorrect cue (right) on the average context maps used on random trials, **(D)** Confidence in inferred context identity following the cue on random trials. * indicates $p < 0.05$, statistical results are detailed in S1 Table.

This degradation of context-specific maps in FI agents had a marked effect on contextual inference - it resulted in a decrease in the inferred probability of the context following the cue on correct trials, indicating a larger chance of inferring the incorrect context (Fig 6D). Therefore, within this modelling framework, we found that increasing environmental overlap between contexts impairs the ability of FI to learn distinct context representations. This deficit can be mitigated by integrating outcome inference during early learning.

## Joint agents outperform alternative implementations

Importantly, this specific implementation of the FI model makes a number of mechanistic assumptions, and so we also tested multiple alternate methods instead of the addition of OI for making inference of a new map easier for FI when there is a large amount of overlap between context features. This allowed us to investigate whether there is an easier way to improve learning in the FI model without the addition of OI. First, we gave the FI agent access to an additional map filled with a random-exploration SR (learnt through random exploration of the environment - see methods) when each new context was first introduced [68–70]. This removes the requirement to create a new, unexplored contextual map upon inference of a contextual switch, which we hypothesised could in some cases decrease the likelihood of switching representations. However, this approach did not improve performance of the FI agent, as the lack of directionality in the random-exploration SR led to a poorer fit with the new experiences than the previous context map, which included directionality of moving along the central stem and then down one of the arms (S10A Fig). Second, we tested the influence of adding the reward as a feature, which could intuitively be thought to further distinguish the two contexts based on reward location [24]. Interestingly, this approach further increased the amount of environmental overlap between different contexts during learning, as the reward prediction on the central stem of the maze is the same between contexts, and thereby unreliably impacted performance (S10B Fig). Lastly, we found that blocking access to the incorrect arm during training, thereby forcing the agent to choose the rewarded arm, a strategy often employed in rodent experiments [14,53,54], also did not consistently improve performance of the feature inference algorithm, suggesting that simply constraining behavioural exploration does not resolve ambiguity in context assignment when posterior separation remains weak (S10C Fig). This was the case, as agents cannot reach the previously rewarded location to unlearn that it is rewarded and must instead gradually learn the new transition structure of the environment. Until they do so, they will continue to attempt to enter the blocked off arm, despite it being blocked. To summarise, within this modelling framework, the most effective way we found to promote separation of context-specific predictive maps in FI was to integrate outcome inference during early learning.

## Performance is robust to training regimen

To investigate whether the benefit of using OI to support FI generalizes to different training regimes beyond blocks of trials consisting of 50 trials each, we investigated the performance of algorithms on (1) different block lengths, where blocks consist of a different number of trials (S11A Fig) and (2) different probabilistic scenarios, where each block instead contained a proportion of trials not from the current trial type, but from the other trial type (S13A Fig). We found that improvements in performance on both block and random trials from combining OI and FI during learning were independent of the block length (S11, S12B and S12C Figs) and the probabilities of the other trial type (S13, S14B and S14C Figs). Our results show that even when trials were not organised into blocks of a single type, outcome inference remained more stable than feature inference because agents could correct their responses within each trial. This stability allowed

the interaction between outcome and feature inference to continue to reduce the number of attempts required at the first reversal (S11, S12, S13, S14C and S14E Figs). Together, these results indicate that OI can learn sufficiently accurate outcome maps within a single experience of the reward, to recognize that a change has occurred when the reward is no longer present in the same location in the next trial.

Interestingly, we found that despite joint inference remaining equally stable across all contingencies, as the block length neared 1 or the probabilities of the other trial type neared 50% the number of OI agents that completed the task within the time limit set on algorithm run time (48hrs per 5 agents) dropped strongly, indicating that decreasing consistency in trial type presentation impaired the ability of OI to consistently learn representations that can support contextual inference (S11, S12, S13 and S14D Figs). These results suggest that having two sources of contextual estimates can stabilize learning, even when the information from both sources is not reliable enough to consistently support learning individually.

Together, our results suggest that interaction of FI and OI must be regulated based on task demands, with OI being important to support learning and inference especially during early learning when FI maps are of poor quality, and OI over-powering well-learnt FI maps preventing within-trial predictive inference, especially within random trials, but also during blocks of trials. Within this paper we manually determine when OI and FI are allowed to interact, however we were curious to see whether the difference in confidence of posteriors from OI and FI would allow for turning OI on when FI maps are poor and turning OI off when FI maps are good. As a preliminary metric, we compared the confidence in OI posteriors and FI posteriors as their average posterior probability over all block trials or over all random trials. As a preliminary test of whether the difference in confidence in posteriors from OI versus FI correlated with the utility of FI, we investigated the impact of placing a simple threshold such that outcome inference would always be turned off to allow for using the well-learnt FI from joint inference on random trials (S15 Fig). We found that this threshold would, firstly, turn on OI during blocks of trials above around 12 distractors around the cue or a cue-choice distance of 12, which we found corresponded well with when the quality of FI maps learnt without the support of OI decreases (S15 Fig). Secondly, this threshold would turn off OI during random trials until around 12 distractors around the cue or a cue-choice distance of 12, where we found that the FI maps became too poor to support within-trial predictive inference and OI algorithms required less attempts per trial (S15 Fig). Indeed, these results suggest that a running average of the confidence in OI and FI could possibly be used to determine whether FI interaction with OI should be turned on or off.

In summary, our results indicate that as overlap increases between contexts, the ability of FI agents to perform within-trial predictive inference decreases. This is similar to previous observations of decreasing performance of RNNs as over-lap between contexts increases in similar tasks [13,29,30]. Interestingly, we found that the deficits in FI could be rescued by using OI to guide initial learning of feature representations. Specifically, we found that OI could be used to stably identify changes in context during learning, allowing for the FI algorithm to learn distinct representations of each context, resulting in long-lasting improvements in contextual inference.

## Joint agents produce internal variables similar to hippocampal splitter cells

A core goal of our study was to frame the interaction between these algorithms in a way that was interpretable from a neuroscience perspective. Neural activity in the CA1 area of the hippocampus (HPC) has been proposed to exhibit an implementation of SRs, where the activity of individual neurons represents the predicted future occupancy of a specific location (S16A Fig) [44,71,72]. Multiple neuroscience studies have shown that a signature of HPC activity in context-dependent tasks is the presence of so called 'splitter cells': cells that fire differently in the same location in an environment depending on the inferred context [73–76]. These cells are thought to allow for similar locations in different contexts to be represented with distinct neural activity [1,35,46]. Splitter cells are often described as prospective, with activity differing according to the upcoming trajectory or behavioural context rather than reflecting only recent history [35], a pattern that has been interpreted as reflecting representation of latent contextual variables [1,73]. We hypothesised that agents utilising FI (therefore both FI and joint inference agents) would produce activity reminiscent of prospective splitter cells on the

central stem of the T-maze after presentation of the cue, as in these models different SRs are learnt and used to guide behaviour in each of the two contexts of the task [1,51]. To investigate this directly, we estimated simulated 'cell firing' in each of our agents, using the future predicted occupancy of a chosen location, weighed by the probability of the context map being inferred (Fig 7D). We then calculated this firing for each context map at every location in the environment to create a set of 'place cells' [44] for each agent. As trials are presented in a randomised order, the past choice is not predictive of the current trial type and therefore any trial type-specific 'cell firing' following the cue represents prospective activity that predicts the choice that will be taken.

We then used this metric to compare simulated cell firing along the shared central arm of the maze - similar to classic 'splitter cell' papers [73–75]. For each context we found the corresponding SR map and modeled simulated cell activity as each 'cell' representing the future predicted occupancy of a specific location on the stem of the maze when the agent is in any other location on the maze (S16A Fig), weighted by the SR maps current posterior probability (Fig 7D). For each simulated cell with a firing field in the central stem, we compared its 'activity' in its preferred context with its 'activity' in its non-preferred context. When we did this, we found that both joint and FI models produced patterns highly reminiscent of 'splitter cells' (Fig 7A and 7B. S16B Fig provides results from Fig 7A using a single sequential colormap). Therefore, similar to previous work [1,21,43,51], in this task FI models can reproduce classic HPC splitter cell experiments.

## Removal of outcome inference mimics loss of HPC splitter cells after PFC lesions

We next examined whether the interaction between FI and OI provides a computational analogue of proposed interactions between prefrontal cortex (PFC) and hippocampus (HPC). In experimental studies, PFC has been implicated in outcome-based contextual inference [17,46,48] and prefrontal input to hippocampus has been shown to influence learning and contextual representation under certain task conditions [15,35–37]. Moreover, if PFC input to the HPC is eliminated, there is a dramatic loss of splitter cells in the HPC [35,46], suggesting that loss of PFC input degrades the representation of context in HPC. In our framework, during learning the joint inference algorithm could be described as using contextual estimates based on OI calculated in PFC, and integrating these with contextual estimates based on feature inference in HPC. We therefore wanted to investigate if removing OI from the joint algorithm during learning could replicate the loss of splitter cells in HPC caused by removal of PFC input. We emphasise that this mapping is intended as a hypothesis-generating interpretation rather than a one-to-one circuit-level model.

To investigate this, we returned to our simulated splitter cell analysis described in Fig 7A and 7B. We compared the properties of simulated cell activity from the FI algorithm with or without the addition of OI. We found that in the full joint inference algorithm cells showed a large difference in firing between their preferred and non-preferred contexts (Fig 7A and 7B). In contrast, in the algorithm without outcome inference, this difference in firing was greatly reduced (Fig 7A and 7B).

We next quantified this loss of splitter cell-like activity more generally by looking at the difference in posterior probabilities that determine the weighting of each cell's firing in its preferred versus non-preferred context, which we refer to as difference in splitter probabilities (Fig 7D). Using this metric, we found large differences in activity in each context along the entire stem of the maze in agents utilising joint inference. Consistent with our previous findings, we also found that these differences were consistently lower in agents without access to OI (Fig 7C). Further, we found that while increasing environmental overlap between contexts had minimal effect on splitter-like activity in joint inference agents, increased overlap led to a large reduction in this difference in activity in agents without OI (Fig 7E and 7F). Again, this degradation was similar irrespective of whether overlap was increased using distractors or distance between the cue and the choice point. Consistent with this, we found that agent-by-agent, decreased behavioural performance without OI was strongly correlated with decreased differences in splitter-like activity (Fig 7E and 7F).

These experiments suggest that the joint inference model provides a framework describing HPC activity during behaviour requiring use of hidden contexts, and also the contribution of PFC to the learning of this behaviour. Using this

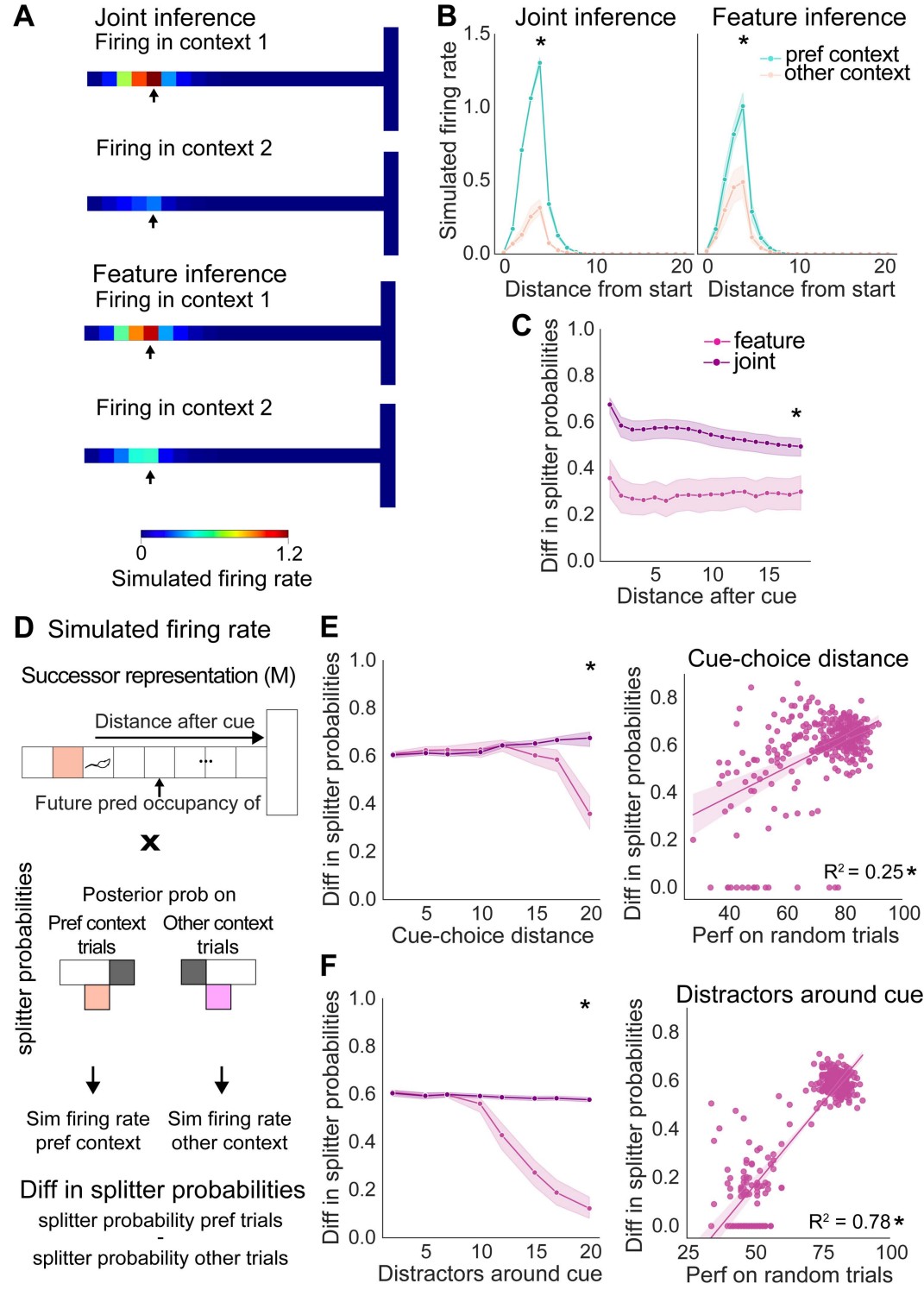

**Fig 7. Removing support from outcome inference during learning mimics experimentally observed splitter cell loss. (A)** Simulated cell firing of a cell representing future predicted occupancy of location 4 (indicated with an arrow) along the overlapping central arm on correct random trials across all agents for cue-choice distance 20, **(B)** Quantification of simulated cell firing along the central arm in its preferred and non-preferred contexts, **(C)** Evolution of the difference in splitter probabilities along the central arm, **(D)** Schematic showing how simulated firing rates are calculated using the successor representation map associated with each context weighted by its posterior probability on its preferred and non-preferred trial types to calculate the future

predicted occupancy of a specific location within all other locations of the environment, **(E)** Impact of increasing cue-choice distance on the difference in splitter probabilities in the location following the cue (left) and correlation between performance on random trials and the difference in splitter probabilities in the location following the cue (right). **(F)** Same as (E) but for distractor features. * indicates p<0.05, statistical results are detailed in S1 Table.

framework, we found evidence to suggest that the strength of splitter cell specificity indicates how different context representations are from one another. In turn, this also indicates how well the agent will perform on tasks involving within-trial predictive inference.

### Outcome inference supports learning of a delayed non-match to sample task

In our results so far, we have shown that supporting FI with OI during learning generates a robust and long lasting basis for performing within-trial predictive inference. Until now however, our experiments have been limited to a relatively simple T-maze task. Although this task is widely used in the neuroscience literature, we next wanted to investigate if the interaction of FI and OI may be a more general mechanism to facilitate performance on contextual inference tasks. To do this, we focused on two alternate tasks - first a classic delayed non match to sample task, which has been used for decades to study PFC and HPC function [77–80], and second a more complex structural learning task, that has been proposed to isolate a core function of the HPC in associative learning [16,81–83].

First, we investigated a delayed non-match to sample task - a behavioural task strongly dependent upon activity in both PFC [15,55,56,80,84] and HPC [77–79,85,86]. In this task, two cues are presented one after the other in time. If the cues do not match (i.e., the first and second cue differ from one another) the agent must make one choice, for example, 'go', or choose right, whereas if the cues match the agent must make a different choice, for example, 'no-go' or choose left (Fig 8A) [55,56]. Therefore, we implemented this task using a similar layout as our T-maze, where two cues were presented one after the other as an agent traversed a central arm. Subsequently, the agent had to make a choice to turn left or right, based on the preceding cues: matching cues (A-A or B-B) cued a right choice, while non matching cues (A-B, or B-A) signalled a left choice. Based on the rodent literature, learning requires that animals are pretrained first on trials associated with only one outcome, before adding in trials associated with the other outcome [15]. To be able to directly compare with this literature, we therefore pretrained agents first on trials where the reward was located on the right arm using blocks of trials of each trial type (i.e., A-A or B-B) (6 blocks each, consisting of 50 trials), followed by training the agent on blocks of all trial types (A-A, B-B, A-B or B-A) (5 blocks each, consisting of 50 trials). Finally, we investigated whether agents learnt to predict the reward location within-trial by testing agents on 500 randomly selected trials.

Based on our previous results, our prediction was that the addition of OI to FI during initial learning would facilitate the performance of FI during and following learning. Therefore, we built a joint algorithm where OI supported FI during the block training phase. We first compared the performance during the block training phase of the joint inference algorithm, where OI supported FI, to the FI algorithm without OI support. Similar to our previous results in the T-maze, we observed that during the delayed non-match to sample task the use of joint inference dramatically improved performance during training (Fig 8B). This improvement also led to improved performance when investigated on random trials. Interestingly however, in contrast to the cued T-maze, the effect was not long lasting, and only apparent for the first 200 trials (Fig 8C). These data suggest that supporting FI with OI during learning also improves the formation of distinct context-representations for context-dependent decision making during a delayed non-match to sample task. Additionally, the overlap between tasks leading to degradation of performance over time when OI was no longer present, indicates that additional support may be necessary to ensure stability of representations after learning. Similarly, despite only being required for the first block switch in the T-maze tasks, here we allowed OI input to support FI for the entire block training phase to allow maximal support throughout training. The degradation observed when OI support was removed is likely due to their being a larger number of tasks that all overlap with one another, resulting in more overlap in general leading to

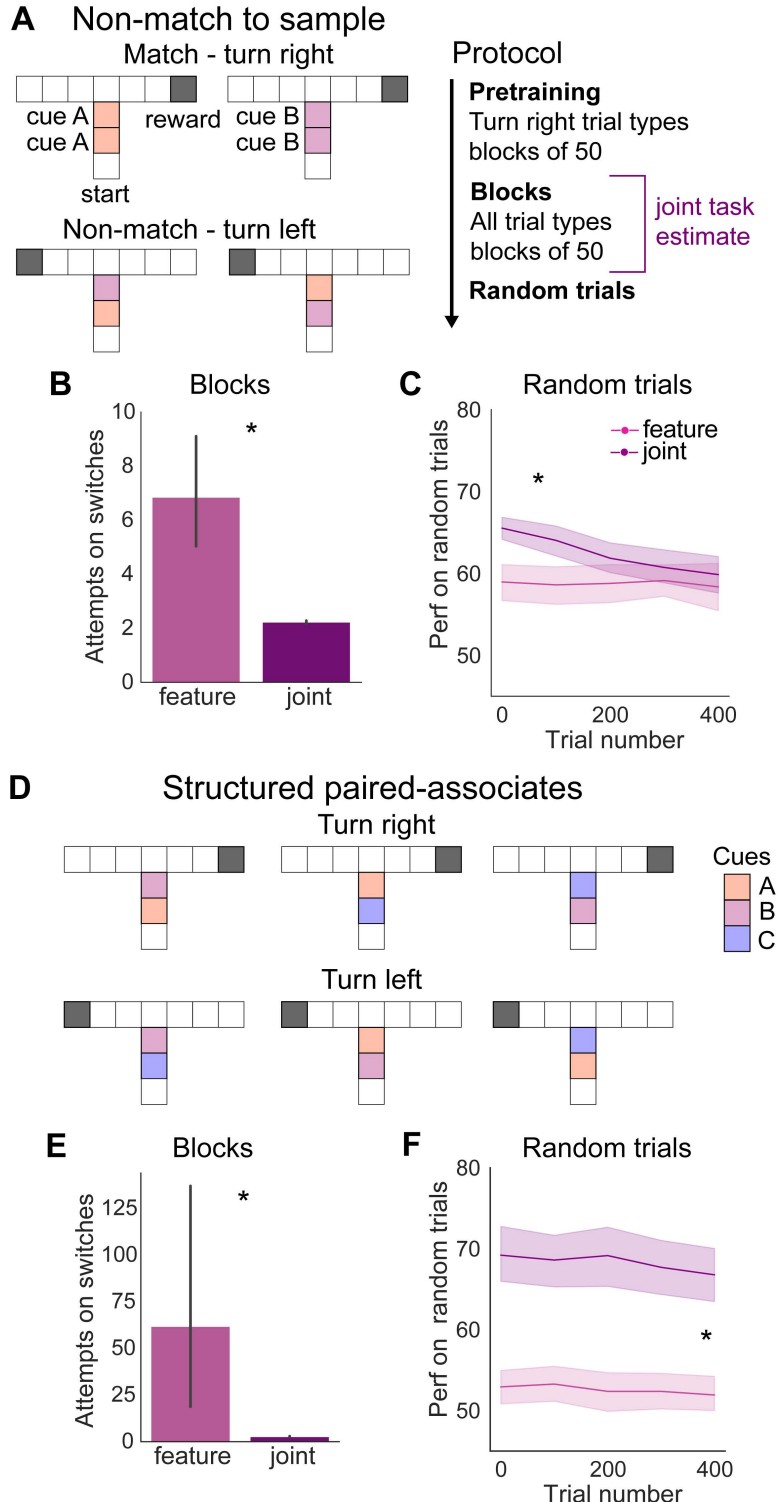

**Fig 8. Supporting feature inference with outcome inference during learning improves performance on cue-discrimination tasks. (A)** Schematic of the non-match to sample task, where trials with the same cue repeated twice require a different response than trials where two differing cues are presented (left) and training protocol showing that outcome inference is used to support feature inference during the training phase (right), **(B)** Performance of feature inference and joint inference on the last 10 block switches, **(C)** Performance of feature inference and joint inference on random trials,

**(D)** Schematic of the biconditional discrimination task, where the meaning of the second cue depends on the identity of the first cue, **(E)** Performance of feature inference and joint inference on the last 10 block switches, **(F)** Performance of feature inference and joint inference on random trials. * indicates p<0.05, statistical results are detailed in S1 Table.

a gradual decay of distinct contextual representations. Notably, this result is highly consistent with data observed in recent animal work, where PFC activity is required for accurate learning of such tasks [15], and further suggests a role of PFC activity may be to provide stable contextual estimates based on outcome inference to HPC as feature representations are being constructed.

### Outcome inference supports learning of a structured paired-associates task

While delayed non-match to sample tasks have been widely historically used, a number of alternate ways to solve such tasks, have resulted in it falling out of favour as a means to study contextual behaviour. Recently developed structured paired-associate discriminations have been designed with the aim to mitigate such issues [16,83]. Much like a delayed non-match to sample task, these tasks consist of trials each with two cues separated in time. However, to prevent the use of alternative strategies, each trial type is formed from a pool of 3 cues (A,B and C) that are combined in a specific way. Firstly, they are ordered such that the meaning of the second cue is dependent on the first (e.g., A-B is different from C-B). Secondly, the identity of the first cue is also uninformative without the second cue (A-B is different from A-C). Finally, the order the cues are experienced in is also critical (A-B is different from B-A). These trial types are arranged into 6 combinations such that the agent must use a combination of the two features in order to solve the task (Fig 8D).

We next investigated if FI agents could solve this complex task, and whether this performance was facilitated by the addition of OI during learning. To test this we trained agents using the same stepwise approach as the non-match to sample task, where we first pretrained in blocks of only 'turn right' cue combinations (3 blocks each, consisting of 50 trials), followed by blocks of both 'turn right' and 'turn left' cue combinations (3 blocks each, consisting of 50 trials) (Fig 8D). In this more complex task, agents relying on FI alone struggled to perform well, plateauing only slightly above chance during the block stage and showing limited generalisation to random trials. Supporting FI with OI during learning improved performance, both during training (Fig 8E), and during random trials (Fig 8F). Interestingly, contrasting with the non-match to sample task, but consistent with the T-maze task, the support provided by OI during learning was essential and led to long lasting improvements on random trials, where agents lacking the support of OI performed poorly, with performance remaining near chance. Together, these results indicate that during learning OI can support the formation of stable FI representations across a range of neuroscience tasks.

## Discussion

This study examined how agents can learn and use contextual information when the features that distinguish contexts are only partially observable. By combining feature-based predictions with outcome-based estimates during learning, our model allows for the acquisition of distinct context representations in both spatial and sequence-learning tasks where sensory evidence for each context is imperfect. This imperfect evidence can be due to sensory information being either unreliable or incomplete. We show that during learning feature-based inference alone becomes unreliable when the SNR of sensory evidence for contexts is low, whereas outcome-based inference remains stable. Our results show that, as a result of its stability in the face of low contextual SNR, outcome-based information can facilitate the formation of stable context-specific maps for feature-based inference during learning. This interaction between feature- and outcome-based inference reproduces several qualitative features of hippocampal and prefrontal function and clarifies the computational roles each may play during learning. The following sections situate these findings within the broader literature on contextual inference, spatial prediction and fronto-hippocampal processing, and outline their limitations and empirical implications.

## Outcome-based contextual inference as a supervisor during learning

The feature-based component of the model struggles when contextual differences are weak because early errors in assigning observations to the correct context distort the learned feature maps in a way that is difficult to reverse. To acquire predictive representations for each context, the agent must both i) infer a distinct context associated with each cue and ii) maintain this belief until the rewarded location is reached to drive appropriate actions. When contextual signal-to-noise is low, however, the feature-based learner continues to interpret observations as arising from the initially inferred context following the first trial-type switch. As a result, the corresponding map is updated with inconsistent information, leading to under-representation of the correct cue and inappropriate incorporation of features from the alternative context. This distortion persists over extended training, reduces confidence in the feature-based posterior following the cue, and produces sustained performance decrements when the context changes (Fig 6).

In contrast, outcome-based inference provides a robust estimate of context during early learning because reward statistics diverge sharply between contexts even when their sensory features overlap. This allows the outcome-based component to generate a clearer posterior precisely at the stage when feature-based maps are most vulnerable to misassignment. Early in learning, posterior confidence from outcome inference therefore exceeds that from feature inference, such that the joint context estimate is dominated by outcome-based information. This joint estimate determines both which context-specific map is used for action selection and which map is updated on each trial, redirecting learning at the first reversal to the appropriate map and allowing distinct predictive representations to emerge. In this sense, outcome inference provides a more reliable context label that effectively supervises feature-map learning. Once feature-based maps are well formed, feature inference operates autonomously and is sufficient to support stable performance. Continued outcome-based interaction beyond this learning phase does not further improve behaviour and can impair performance when trial types are unpredictably interleaved, indicating that the FI–OI interaction is phase- and task-dependent rather than continuously stabilising.

Because outcome-based inference updates context primarily following unexpected reward or omission of expected reward, its ability to support feature learning depends on both within-trial and across-trial structure. We find that its benefit during learning persists even when trials are alternating (S11 and S12 Figs) or randomised (S13 and S14 Figs). In the tasks studied here, this robustness arises from the within-trial structure: although agents make only a single choice per attempt, they are allowed repeated attempts until the correct outcome is obtained. Under these conditions, outcome inference can rapidly accumulate sufficient evidence following a trial type change to stabilise feature-map learning, even when mismatched reward statistics are sampled only once, consistent with prior work showing benefits of outcome inference when agents are allowed to explore until reward is found [51].

Although block structure is often imposed experimentally because it facilitates learning in rodents and humans [9,10,46,62–64], periods of stable contingencies are also common in natural environments, where reward locations, environmental conditions, or behavioural goals persist over extended intervals. For example, repeated foraging from a single food source or navigation along a familiar route both generate sustained contextual stability [87]. Together, these results indicate that outcome-based support is most effective when contextual contingencies persist long enough for reward statistics to be estimated with reasonable confidence. They also suggest routes for extending the framework to more highly interleaved environments, for example through replay of experiences at the end of trials once reward statistics have been reliably inferred, allowing learning to be reinforced within the appropriate feature map [51,88–91].

An interesting observation emerges from comparing the OI–FI interaction in more complex paired-associate style tasks (Fig 8). Although these tasks share similar relational rules, they behave differently. As in the core T-maze, sustained outcome-based influence is not required in the structured paired-associate task once learning has stabilised, whereas performance in the biconditional discrimination task gradually degrades in its absence. While both tasks involve overlapping feature structure, they differ in their temporal organisation. The structured task imposes sequential ordering constraints that may promote more stable predictive trajectories once context-specific maps are formed. In contrast, the

biconditional task relies on often symmetric cue combinations without explicit directional progression, which may render context estimates more susceptible to gradual misassignment over extended training. Under such conditions, intermittent outcome-based recalibration may help preserve stability. Together, these observations suggest that task structure can strongly influence how feature- and outcome-based inference interact.

**Feature inference as a model of the hippocampus: contextual inference under partial observability**

The idea that the hippocampus supports feature-based inference for decision making builds on long-standing theories of hippocampal function, particularly in the context of tasks that require contextual inference.

Specifically, the hippocampal involvement has been reported in context-dependent decision making when information relevant for action selection is encountered prior to, but not at, the moment of choice [14,53,77–79,86,92], or information is present at the moment of choice, but is obscured by other features [11,12]. In such situations, successful performance may require drawing on internal representations of the recent past to resolve ambiguity in the present. This form of partial observability can arise for two main reasons. First, relevant sensory features may be separated from the decision point in time or space, so that the immediate sensory input provides only an incomplete description of the current context. Second, different contexts may share highly similar sensory features, obscuring the relevant sensory information. Both forms of partial observability are common in rodent navigation and working-memory tasks, which have been traditionally used to study hippocampal contextual representations and remapping [1,11,12,93,94]. Both kinds of partial observability are present in the cued T-maze considered here: the cue that specifies the rewarded arm is encountered before the decision point and the sensory features available along the maze stem are largely shared across contexts. Similarly, within the sequence-dependent tasks, additional partial observability stems from the specific identity and order of multiple observations that occur at different locations or times.

In the cued T-maze, this separation between the cue and the decision point means that the sensory information available at the moment of the primary choice is insufficient to determine the correct action. The agent encounters the contextual cue only at the start of the central stem; by the time it must choose which arm to enter, the cue is no longer visible and the local sensory environment is identical across contexts. Successful performance therefore requires maintaining an internal representation of the context inferred from earlier observations and integrating this with the animal's current position. This form of inference is closely related to long-standing ideas about hippocampal coding, in which contextual representations are constructed from recent sensory history and are updated as the statistics of the environment change [1,73]. It also clarifies why tasks that preserve the cue at the decision point can be solved without the use of contextual inference: if the relevant information is directly observable when the choice is made, the mapping to the correct action does not depend on a latent contextual variable, as seen in cued tasks where hippocampal activity is dispensable [60,95,96], in contrast to conditions where the cue and choice are spatially or temporally separated [14,53,77–79,86].

In the present framework, context inference determines which context-specific TD map is active, and this map in turn governs action selection at each state. Although contextual evidence is typically acquired at cue onset, the decisive action at the junction is generated by the currently active policy rather than by a fixed commitment made at cue presentation. In this sense, the model is hierarchical: a latent context estimate selects a policy, and that policy is executed across states unless updated by new evidence.

Taken together, the framework suggests a division of labour that depends on task structure. When contextual cues remain available at the decision point and discriminability is high, behaviour can be supported by direct cue–response policies with minimal reliance on predictive mapping. When cues are temporally displaced from choice or obscured by other features, successful performance requires maintaining or reconstructing contextual state, a regime in which hippocampal-like predictive representations are expected to contribute. Finally, when contextual features are weak or highly overlapping, outcome-based contextual labelling becomes more reliable than feature-based inference, increasing the expected contribution of prefrontal-like outcome signals, particularly during learning or under persistent ambiguity.

The tasks used here build on contextual inference paradigms where a single action is based on a context defined by a single feature or a set of independent features, like a single cue or past outcome [4,9]. In contrast, the spatial cued T-maze used here involves executing a sequence of actions to traverse space, with the decisive choice occurring at the junction, rather than a single action, and the sequential cued T-maze used here relies on interpreting multiple interdependent cues based on their order and cue identity, rather than a set of independent features. Importantly, these structural differences do not imply that such tasks are uniformly hippocampal dependent; rather, they allow us to test how context inference mechanisms operate under increased sequential and spatial demands. Computationally, this requires maintaining and applying contextual information across a sequence of actions linking early observations to the eventual decision [1,10,51]. Successor-like representations found in the hippocampus capture these structural and predictive relationships [24,44,51] and therefore provide a natural substrate for contextual inference in tasks that depend on sequential observations. The cued T-maze can thus be understood as a multi-step, spatial instance of simpler contextual inference problem, where effective behaviour depends on combining past sensory events with an internal model of how those events unfold across space. Similarly, the sequential T-maze can be understood as a sequential instance of simpler contextual inference problem, where effective behaviour depends on combining a sequence of past sensory events with an internal model of how those events unfold across space or time. Together, this supports the idea that the hippocampus relies on detailed, predictive representations like successor representations to support contextual inference.

Consistent with this interpretation, the feature-based component of the model exhibits representational properties that resemble several phenomena reported in hippocampal recordings (Fig 7), including the formation of distinct context-specific maps and the capacity to update these maps when the statistics of the environment change. In many accounts of hippocampal function, spatial and non-spatial features are integrated into structured representations that support navigation, memory and contextual discrimination, and these representations have been described as reorganising when sensory inputs diverge sufficiently from prior expectations [1,24,94]. Within this framework, remapping reflects an inference process in which deviations from predicted sensory features indicate that a different context is active. The feature-based learner implements a mechanistic analogue of this process: it constructs predictive state representations for each context, and it reallocates observations when they no longer accord with the transition structure of the currently inferred map. This provides a direct account of how distinct hippocampal maps can arise both during structured tasks and in settings such as free foraging, where changes in environmental statistics are sufficient to trigger a reorganisation of place-cell activity.

### Relationship to hippocampal and prefrontal computations

In contrast, the outcome-based component reflects computations more closely associated with prefrontal regions that track task structure, behavioural rules and the consequences of actions over time. Medial prefrontal cortex, orbitofrontal cortex and anterior cingulate cortex are all implicated in maintaining information across delays and trials, integrating recent outcomes to form abstract representations of latent contexts [46,51,97,98]. These processes parallel the role of the outcome-based learner, which summarises recent reward structure into a compact contextual estimate that remains reliable even when sensory features provide an ambiguous guide to context. By extracting regularities from the pattern of outcomes rather than relying on detailed sensory predictions, this component offers a complementary route to contextual inference that is consistent with the functions typically attributed to frontal circuits and their interactions with the hippocampus.

The interaction between the feature-based and outcome-based components also captures several qualitative features of hippocampal activity observed during contextual navigation, including the emergence of splitter-like coding and its dependence on prefrontal input. In the model, distinct feature-based maps give rise to context-specific firing patterns along the cue-free portion of the maze stem, producing context-dependent modulation of predicted future occupancy along cue-free stem locations; this resembles prospective differences in activity, but in our model these differences arise from maintained context estimates rather than from an explicit commitment to a future choice. This mirrors the prospective

component of splitter cells reported in the hippocampus, which depends on long-range interactions with medial prefrontal cortex via nucleus reuniens [35]. Within our model, when outcome-based support is removed during learning, the model no longer forms stable context-specific feature maps and this prospective activity following contextual cues is markedly reduced, consistent with the loss of splitter signalling following perturbations of prefrontal or reuniens input [35]. In contrast with our model's predictions, this experimentally observed loss of splitter signalling within the hippocampus occurred with inhibition of prefrontal inputs after animals were trained on a task, rather than during learning. However, although the task used in the experiments was a T-maze, it did not contain an explicit sensory cue, but rather the behaviour depended on the outcome of the previous trial. Under these conditions, hippocampal involvement is reduced, and disruption of hippocampal splitter activity has been reported not to impair behavioural performance in this specific task variant [35]. This effect can thus be explained by our model, as removing outcome inference input to feature inference during a task where outcome inference is the only algorithm able to infer the contexts, would lead to loss of splitter cells in feature inference independent of the training stage, as has been shown in [51]. This is the case, as feature inference can only perform contextual inference when the context can be inferred based on sensory features within the environment, which is not the case within this task. We provide detailed predictions on when splitter cell loss would be expected in distinct tasks in our experimental predictions section below.

Our framework therefore links the emergence of context-sensitive coding to the successful acquisition of distinct predictive representations, and to the broader circuit motifs thought to sustain communication between hippocampus and frontal cortex during tasks that require inference under partial observability.

## Relationship to existing models of contextual inference

The FI–OI framework can be situated within a wider ecosystem of models that address contextual inference under partial observability. The outcome inference and feature inference models used here build on Bayesian approaches to latent-state inference, which characterise how agents combine sensory observations and reward history to infer hidden contexts under partial observability [1,4,9]. Bayesian latent-state approaches rely on explicit definition of the prior defining how observations relate to different contexts, allowing for incorporation of context persistence or block structure [76, 4, 9]. However, as a result, simple latent-state inference approaches are limited to (1) choosing a single action and (2) inferring contexts defined by a single feature or a combination of independent features in a pre-defined manner, like a single cue or past outcomes. Such models have been applied to situations in which relevant information is separated in time from the moment of choice or where multiple contexts share overlapping sensory features, and they capture the increased difficulty of inference when contextual SNR is low [1,4,9]. Rather than proposing a fundamentally different inference principle, our model removes explicit assumptions of how observations relate to context from the prior, instead learning about all feature- and outcome-based information. This approach allows us to separate the roll of feature- and outcome-based information in contextual inference, so that we can analyse the conditions under which each inference approach succeeds or fails.

The present framework incorporates the ideas of bayesian latent-state inference but extends them to multi-step environments, where agents must link cues encountered early in a trajectory to decisions made several states later. The feature and outcome inference model used here expand on these Bayesian approaches by incorporating explicit models of the world, allowing them to be applied to complex problems that involve determining trajectories consisting of multiple actions. Specifically, the feature inference model incorporates successor representations, that learn a predictive model of how all of the features of the world are related. Successor representations naturally capture the multi-step structure of spatial or abstract environments allowing for planning of multi-step trajectories to reach rewards [44,51,70,91]. As a result our framework could extend beyond T-mazes to more complex grid-based environments [51]. In addition to supporting learning of multi-step trajectories, successor representations can be used to quickly update trajectories when the reward location changes, although changes in transition structure require updating of the successor representation itself, resulting in slow updating of behaviour [70,91].

Extending successor representations with Bayesian inference frameworks in feature inference allows for incorporating context-dependent structure by assigning separate predictive maps to different environmental contingencies [24]. As a result, feature inference allows information encountered early in a trajectory to influence predictions and thereby selected actions several states ahead, providing a principled mechanism for linking features with the long-range consequences of actions [24]. This allows for inference of contexts based on multiple inter-dependent features at different locations of the environment, allowing for the solving of tasks depending on sequences of features. Related principles appear in graph-structured contextual inference schemes, which combine a different kind of model-based representation with Bayesian inference [26]. Contrastingly, extending outcome-based models allows for incorporating context-dependent structure by assigning separate predictive maps to different trajectories and associated behavioural outcomes [51].

Our present framework clarifies the limits of the predictive mapping strategy of feature inference under partial observability: when sensory evidence provides only weak discrimination between contexts, the prediction errors that drive map separation become too small to support reliable contextual inference. By combining successor-like predictions with outcome-based contextual estimates, our joint model reveals how stable context-specific representations can nonetheless be acquired, and how predictive maps may interact with outcome-based signals during learning in environments where sensory ambiguity is substantial. Our results show that outcome-based information can compensate for this low signal-to-noise ratio by providing a more stable source of contextual evidence during learning, enabling distinct representations to form even when sensory overlap is substantial.

Recurrent neural network models offer a complementary account of contextual inference, learning latent structure using recurrency instead of Bayesian contextual inference and using function approximation based on all sensory information, including features and outcomes, instead of learning models based on explicit learning rules with specific focuses on features or outcomes. Recurrent neural network models can acquire internal representations that support context discrimination in partially observable environments [10,13,21–23], have been used to model the hippocampus [10] and also struggle to learn contextual representations when the contextual SNR is low [9,10,13,23,29,30]. While these approaches demonstrate that context representations can emerge through end-to-end optimisation, the resulting internal dynamics are often difficult to interpret and do not reveal which specific computations support successful inference [99–101]. By contrast, the present framework separates feature-based and outcome-based contributions, making explicit the circumstances in which each mechanism fails and how their interaction stabilises context-specific maps during learning. This decomposition provides a complementary perspective on the circuit-level computations that recurrent models implicitly capture, and generates experimentally testable predictions about when additional outcome-based input should be required to support reliable contextual inference.

More broadly, our model falls under the category of hierarchical reinforcement learning, which involves grouping multiple state-action pairs together into a single component - often referred to as an option - and then generating a policy based on these higher-level options. Traditionally, hierarchical reinforcement learning models group together multiple state-action pairs that allow for reaching subgoals within the environment to overcome limitations associated with having a large state space, allowing for more efficient initial exploration of the space, as well as faster learning if the structure of the environment or reward location changes [102,103]. Within our implementation, the options can be viewed as the policies defined by each context-dependent TD map (i.e., turn left vs turn right), however, contrasting with traditional hierarchical models, the separation and use of options is gated through the current inferred latent state. As a result, the hierarchical nature of our model is used for learning and activating distinct policies depending on the context, rather than allowing for better learning within a single context, as is the case in traditional hierarchical reinforcement learning.

## Limitations and scope of the framework

The framework presented here is deliberately abstract and is intended to capture the computational requirements of contextual inference rather than the detailed neural mechanisms that implement it. The model relies on a discretised

representation of space and assumes that inference occurs continuously as the agent moves from one state to the next. These assumptions simplify the underlying dynamics and avoid strong commitments about when decisions are taken, how sensory information is maintained over time and space or how transitions between internal states are realised biologically. In practice, the relevant computations are likely to be distributed across multiple brain regions and operate over a mixture of discrete event-triggered updates (e.g., at cue onset or outcome) and more continuous processes. For example, it is important to note that the model used here assumes that the inferred context is updated throughout each trial and that action selection at the junction depends on a maintained context estimate. This continuous updating is a simplifying computational assumption; biologically, context updating may be more strongly driven by salient events such as cue onset, outcome feedback, or unexpected transitions. Related to this, in simple cue-guided tasks, behaviour could in principle be supported by committing to a motor plan at cue onset without further inference. Our aim is not to adjudicate between these strategies in the basic T-maze, but to provide a framework for understanding how latent contextual information can be maintained and applied under conditions of sensory ambiguity. This distinction becomes particularly relevant in more complex relational tasks, where sequential integration and updating of context are less easily reduced to single cue-triggered policies. The model therefore provides only a coarse description of how contextual information might be propagated along a trajectory and should be interpreted as a schematic account of the computational problem rather than a literal account of neural implementation. In the future, our model could be fit to both behavioural and neural data from the hippocampus, prefrontal cortex and thalamus, to establish how these abstract parameters are biologically defined and implemented.

Similarly, the way in which the outcome and feature inference agents learn about the environment and identify different contexts depends on the initialization of parameters for the algorithms. As a result, different behavioural tasks will require the use of distinct parameters. For example, for outcome inference, the filter length determines how distinct different reward locations need to be to be recognised as belonging to distinct contexts. For both algorithms, the optimal prior will depend on the number of distinct contexts. Despite this, it is worth noting that these parameters are all interpretable from a computational and neuroscience perspective, and remain far fewer than those needed to fit an equivalent neural network approach. These constraints are useful for understanding the boundary conditions under which feature-based inference breaks down and how outcome inference can be used to support feature inference, but they leave open how biological systems adjust their internal models or specifics of their contextual inference to cope with different kinds of tasks. Future extensions using hierarchical models [26,49] or richer forms of predictive representations [100,101] may therefore be required to capture the full flexibility of hippocampal–prefrontal circuits that supports learning of multiple different kinds of tasks.

Additionally, in the current implementation of our model, we manually determine when outcome inference should be used to support feature inference - by manually defining this interaction for the first 200 trials. In the future, it would be interesting to see how this can be automated, for example, by weighing the current utility of outcome versus feature inference. One way to achieve this could be to use the discrepancies in predictions between each model, as recently proposed for combining model-based and temporal difference learning to solve contextual inference problems [52]. We provide preliminary evidence that the average confidence in contextual estimates provided by feature inference and outcome inference could be used to decide whether joint inference should or should not be used. Our results indicate that a single threshold on confidence in feature -versus outcome-based estimates could allow for turning on joint inference during blocks of trials if the feature inference map has not been learnt well and turning off joint inference during random trials if the feature inference map has been learnt well. This addition would allow for investigation of how the brain may determine when it is necessary to provide support to feature-based approaches using outcome-based information.

## Experimental predictions

The model makes a set of experimentally testable predictions about when hippocampal representations depend on outcome-based input from prefrontal cortex, and how this dependence varies across learning and task structure.

First, disrupting prefrontal input to the hippocampus, either directly or via nucleus reuniens, should affect behaviour only when two conditions are met: the task context must be partially observable, and the context must be predictively inferable within a trial from sensory features. When these conditions are not satisfied, disruption of prefrontal inputs is not predicted to impair behaviour. If the context is fully observable at the time of choice, hippocampal representations can be learned and expressed independently of prefrontal input, and perturbations should have little effect on either behaviour or hippocampal activity. Conversely, if the context cannot be inferred from within-trial sensory features and instead depends on behavioural outcomes from previous trials, disruption of prefrontal input to the hippocampus should abolish prospective context-dependent coding in the hippocampus without affecting behavioural performance [35,51].

When a task is both partially observable and predictively inferable within a trial, the model predicts distinct effects of prefrontal perturbation during versus after learning. Disrupting prefrontal input during learning should impair behavioural performance and reduce the emergence of prospective coding along cue-free portions of a trajectory, reflecting a failure to acquire distinct context-dependent hippocampal maps. The magnitude of this impairment should scale with the degree of partial observability, such that increasing the distance or delay between contextual cues and behavioural outcomes increases reliance on prefrontal input. Accordingly, graded reductions in cue reliability should produce graded increases in the behavioural and neural consequences of prefrontal disruption. Conversely, enhancing prefrontal input during learning is predicted to improve context discrimination and strengthen prospective hippocampal coding under conditions of low sensory signal-to-noise.

After learning, the model predicts the opposite pattern. Disrupting prefrontal input should improve performance and strengthen prospective hippocampal coding by preventing interference from outcome-based strategies once stable feature-based representations have been acquired. Similarly, artificially enhancing prefrontal input after learning should impair performance and reduce prospective coding, independent of the degree of partial observability.

Together, these predictions delineate the conditions under which hippocampal representations require outcome-based support and provide a framework for dissociating learning-related and performance-related roles of hippocampal–prefrontal interaction. While the present work focuses on prefrontal contributions to hippocampal learning, the framework also predicts reciprocal interactions, such that feature-based inference can stabilise outcome-based learning in environments with volatile or weakly structured trial statistics.

## Conclusion

Together, these results provide a computational account of how contextual inference can emerge from the interaction between feature-based predictions and outcome-based estimates when sensory information is either temporally displaced from action or only partially informative. By extending latent-state frameworks to multi-step spatial and sequence-based tasks using successor representations, the model clarifies when feature-based mechanisms alone become unreliable and how outcome-based information can stabilise the formation of distinct context-specific representations under conditions of high sensory overlap. This division between feature- and outcome-based inference offers a principled interpretation of how hippocampal and prefrontal circuits may contribute complementary evidence during learning, particularly when cues are transient or separated from the point of choice.

More broadly, the model highlights how combining predictive mapping with outcome-based structure learning provides a flexible solution to contextual inference, and offers a general framework for understanding how biological networks integrate multiple sources of information to guide behaviour in partially observable environments.

## Supporting information

**S1 Fig. Schematics of model designs.** (A) Schematic of HPC model, showing how observed feature transitions are compared with existing successor feature maps using Bayesian inference to determine which context is currently most likely, (B) Schematic of PFC model showing how observed convolved reward is compared with existing maps of convolved

reward using Bayesian inference to determine which context is currently most likely, (C) Schematic of joint model architecture, showing how the context likelihoods generated by both HPC and PFC models are combined for contextual inference and action selection.
(TIF)

**S2 Fig. Inconsistent performance of algorithms using other ways of combining outcome inference and feature inference during learning.** Performance on random trials with increasing distractors around cue or increasing distance from cue to choice using (A) replay of trial observations into the context map inferred by outcome inference at the end of trials, (B) replacing the outcome inference algorithm with an ideal observer model. * indicates $p < 0.05$, statistical results are detailed in S1 Table.
(TIF)

**S3 Fig. Performance of algorithms across learning during blocks of trials.** Number of attempts required to make the correct choice on each trial across blocks of trials for different numbers of distractor features (left to right) for FI, OI and joint algorithms (top to bottom). * indicates $p < 0.05$, statistical results are detailed in S1 Table.
(TIF)

**S4 Fig. Performance of algorithms across learning during blocks of trials.** Number of attempts required to make the correct choice on each trial across blocks of trials for different cue-choice distances (left to right) for FI, OI and joint algorithms (top to bottom). * indicates $p < 0.05$, statistical results are detailed in S1 Table.
(TIF)

**S5 Fig. Evolution of performance across blocks highlights importance of first block switch.** Number of attempts required to make the correct choice on each trial (A,B) directly following a block switch or (A,C) within a block for different numbers of distractor features (left to right) for FI, OI and joint algorithms (top to bottom). * indicates $p < 0.05$, statistical results are detailed in S1 Table.
(TIF)

**S6 Fig. Evolution of performance across blocks highlights importance of first block switch.** Number of attempts required to make the correct choice on each trial (A) directly following a block switch or (B) within a block for different cue-choice distances (left to right) for FI, OI and joint algorithms (top to bottom). * indicates $p < 0.05$, statistical results are detailed in S1 Table.
(TIF)

**S7 Fig. Joint task estimates are vital on first block switch and detrimental if used during random trials.** Number of attempts required on last 500 blocks of trials (left) and percent correct on last 100 random trials (right), depending on how long a joint task estimate is used for inference, for distractors around cue (A) and cue-choice distance (B). * indicates $p < 0.05$, statistical results are detailed in S1 Table.
(TIF)

**S8 Fig. OI posterior probabilities on first block switch drive alignment of FI posterior probabilities in joint inference.** Average posterior probabilities across locations of the FI and OI maps representing the previous (now incorrect) trial type following the first attempt on the first (left) and last (right) block switch, in individual (left) and joint (right) algorithms, across increasing distractor features (top to bottom). * indicates $p < 0.05$, statistical results are detailed in S1 Table.
(TIF)

**S9 Fig. OI posterior probabilities on first block switch drive alignment of FI posterior probabilities in joint inference.** Average posterior probabilities across locations of the FI and OI maps representing the previous (now incorrect) trial

type following the first attempt on the first (left) and last (right) block switch, in individual (left) and joint (right) algorithms, across increasing cue-choice distances (top to bottom). * indicates $p < 0.05$, statistical results are detailed in S1 Table.
(TIF)

**S10 Fig. Inconsistent performance of algorithms using methods other than outcome-inference to support learning.** Schematics of effects on performance (top) and effects on performance on random trials (bottom) with increasing distractors around cue or increasing distance from cue to choice using (A) pre-exploration of the environment without rewards or predictive cues, (B) an additional feature that indicates reward presence (the data in cue-choice distance is incomplete as individual agents took over the 48hr time limit to run), (C) forced choice where the incorrect arm is blocked off during learning. * indicates $p < 0.05$, statistical results are detailed in S1 Table.
(TIF)

**S11 Fig. Joint inference is beneficial even when trained on blocks consisting of individual trials.** (A) Schematic of different block lengths tested, (B) Performance on random trials and (C) block trials with increasing distractor features depending on block length, (D) Proportion of agents that completed the task within the runtime limit (48hrs per 5 agents), (E) Number of attempts required to make the correct choice on each trial directly following a block switch (top) or within a block (bottom) for 20 distractors. * indicates $p < 0.05$, statistical results are detailed in S1 Table.
(TIF)

**S12 Fig. Joint inference is beneficial even when trained on blocks consisting of individual trials.** (A) Schematic of different block lengths tested, (B) Performance on random trials and (C) block trials with increasing cue-choice distance depending on block length, (D) Proportion of agents that completed the task within the runtime limit (48hrs per 5 agents), (E) Number of attempts required to make the correct choice on each trial directly following a block switch (top) or within a block (bottom) for cue-choice distance 20. * indicates $p < 0.05$, statistical results are detailed in S1 Table.
(TIF)

**S13 Fig. Joint inference is beneficial even when trained on random probabilities.** (A) Schematic of different probabilistic setups tested, (B) Performance on random trials and (C) block trials with increasing distractor features depending on block probabilities, (D) Proportion of agents that completed the task within the runtime limit (48hrs per 5 agents), (E) Number of attempts required to make the correct choice on each trial directly following a block switch (top) or within a block (bottom) for 20 distractors. * indicates $p < 0.05$, statistical results are detailed in S1 Table.
(TIF)

**S14 Fig. Joint inference is beneficial even when trained on random probabilities.** (A) Schematic of different probabilistic setups tested, (B) Performance on random trials and (C) block trials with increasing cue-choice distance depending on block probabilities, (D) Proportion of agents that completed the task within the runtime limit (48hrs per 5 agents), (E) Number of attempts required to make the correct choice on each trial directly following a block switch (top) or within a block (bottom) for cue-choice distance 20. * indicates $p < 0.05$, statistical results are detailed in S1 Table.
(TIF)

**S15 Fig. Average confidence in estimates allows for selecting whether to turn OI on or off.** Difference in confidence between outcome inference estimate and feature inference estimate across random and block trials, allowing for thresholding when to use joint inference and when to use feature inference (top). Number of attempts on blocks of trials (middle) and number of attempts and performance on random trials (bottom) across increasing distractors (left) and cue-choice distance (right), showing when the selected threshold would result in use of joint inference rather than feature inference. * indicates $p < 0.05$, statistical results are detailed in S1 Table.
(TIF)

**S16 Fig. Supporting information about simulated firing rates.** (A) Schematic showing the distinction in plots of successor representations used in Figs 3 and 6 to investigate the representations learnt by agents looking at the predicted future occupancy of all other features given the current location (pink) and the plots used to show simulated firing rates in Fig 7, which represent the future predicted occupancy of a specific location across all other features (blue), (B) Simulated firing rate maps from Fig 7A originally plotted using the traditional colormap for place cells, allowing for direct comparison with [35], in a single sequential colormap instead.
(TIF)

**S1 Table. Statistical results.**
(PDF)

# Acknowledgments

We thank Daniel Bush, Jesse Geerts, Neil Burgess and other members of the MacAskill and Burgess labs for helpful discussions and feedback. The authors acknowledge the use of the UCL Kathleen High Performance Computing Facility (Kathleen@UCL), the UCL Myriad High Performance Computing Facility (Myriad@UCL), the Edinburgh Compute and Data Facility (ECDF) (http://www.ecdf.ed.ac.uk/) and associated support services, in the completion of this work.

# Author contributions

**Conceptualization:** Jessica Passlack, Andrew MacAskill.

**Data curation:** Jessica Passlack.

**Formal analysis:** Jessica Passlack.

**Funding acquisition:** Jessica Passlack, Andrew MacAskill.

**Investigation:** Jessica Passlack.

**Methodology:** Jessica Passlack.

**Resources:** Jessica Passlack.

**Software:** Jessica Passlack.

**Supervision:** Andrew MacAskill.

**Validation:** Jessica Passlack.

**Visualization:** Jessica Passlack.

**Writing – original draft:** Jessica Passlack.

**Writing – review & editing:** Jessica Passlack, Andrew MacAskill.

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
