## [Decision Letter · Decision Letter 0]

14 Oct 2025

Contextual inference through flexible integration of environmental features and behavioural outcomes

PLOS Computational Biology

Dear Dr. MacAskill,

Thank you for submitting your manuscript to PLOS Computational Biology. After careful consideration, we feel that it has merit but does not fully meet PLOS Computational Biology's publication criteria as it currently stands. Therefore, we invite you to submit a revised version of the manuscript that addresses the points raised during the review process.

Please submit your revised manuscript within 60 days Dec 14 2025 11:59PM. If you will need more time than this to complete your revisions, please reply to this message or contact the journal office at ploscompbiol@plos.org. Please include the following items when submitting your revised manuscript:

We look forward to receiving your revised manuscript.

Kind regards,

Alejandro Tabas, Ph.D.

Academic Editor

PLOS Computational Biology

Daniele Marinazzo

Section Editor

PLOS Computational Biology

**Journal Requirements:**

3) We notice that your supplementary Figures are included in the manuscript file. Please remove them and upload them with the file type 'Supporting Information'. Please ensure that each Supporting Information file has a legend listed in the manuscript after the references list.

4) When completing the data availability statement of the submission form, you indicated that you will make your data available on acceptance. We strongly recommend all authors decide on a data sharing plan before acceptance, as the process can be lengthy and hold up publication timelines. Please note that, though access restrictions are acceptable now, your entire data will need to be made freely accessible if your manuscript is accepted for publication. This policy applies to all data except where public deposition would breach compliance with the protocol approved by your research ethics board. If you are unable to adhere to our open data policy, please kindly revise your statement to explain your reasoning and we will seek the editor's input on an exemption. Please be assured that, once you have provided your new statement, the assessment of your exemption will not hold up the peer review process.

**Reviewers' comments:**

Reviewer's Responses to Questions

**Comments to the Authors:**

Reviewer #1: Summary:

This work presents a model of contextual inference which combines feature-based and outcome-based inference. The authors demonstrate that this joint model succeeds in inferring context even in low SNR situations, while the feature-based inference model alone fails as the contexts become more similar. They connect these two distinct forms of inference to the hippocampus and prefrontal cortex, and demonstrate that inactivating the outcome-based inference inhibits the formation of hippocampal “splitter” cells, just as inactivating PFC inhibits their formation experimentally. The work provides a novel mechanistic framework for understanding PFC-hippocampal interactions during context-dependent learning. This is an interesting model and story. However, there are some additional considerations about when and why this joint model provides a benefit that are required.

Major Points:

1. I would like to see the authors examine in more detail the dependence of learning on the “blocked” trial structure. I think it would be valuable to see the model’s learning performance for random trials, interleaved trials, and blocked with varying levels of added stochasticity. Critically, the lack of these tests makes it a bit vague as to what exactly is necessary for outcome inference to have a benefit. Further, since naturalistic behaviors are not so strictly blocked, and the authors propose that HPC and PFC act as feature and outcome inference, then it should generalize to non-blocked (or at least less strictly blocked) trial structures. This is my biggest sticking point. I would like to see some sort of investigation into this and more discussion about how important blocking is for outcome inference. It doesn’t need to be a sweep over all sorts of parameters in the trial structure, but I would like to see the failure in a couple common non-blocked trial structures, and also see how it approaches that failure (is it sudden or gradual?).

2. Paper was at times difficult to follow, I’ll include specific details in my minor points section. Figures could also use some cleaning up, there is text of different sizes in some places, lines are a bit thin and hard to read. No units in colorbar of Figure 6A. Also, should use a single continuous colormap for 6A as opposed to a tricolor one, as firing rate is a continuous variable and the transition areas of the current colormap (.4 and .8) are arbitrary. .3 vs .5 should look like the same difference as 0 to 0.1.

3. I think it would be important to show performance curves during the training stage, again for different trial structures as in point 1.

4. The results for Figure 5-S1B and Figure 5-S1C are strange to me. How would adding a reward (in a different location) increase environmental overlap? Further, if the agent is forced into the correct arm of the maze, the SR for each context must be different, so why didn’t this improve performance of the FI algorithm? I think these need a bit more investigation and explanation.

5. This work should be better contextualised. The discussion/introduction should be extended to include discussion on working memory/contextual studies of hippocampal function, as these appear directly relevant. A few examples: I. Lee and R. P. Kesner. Nature neuroscience, 2002. ; Lee and Kesner. Behavioral neuroscience, 2003.; Gilbert and Kesner. Behavioural brain research, 2006.

There is also recent RL contextual modelling work of hippocampal function that appears related to the present work and thus relevant to discuss and contrast (Pedamonti et al. BioRxiv: Hippocampus facilitates reinforcement learning under partial observability).

Minor Points:

1. From a writing perspective, the flow can be a bit choppy. I will give a couple of examples, but it exists throughout the text. In the abstract, “the ability to use the context we are in to flexibly adjust our decision-making is vital for navigating a complex world” probably reads better simply as “the ability to use context to flexibly adjust decision-making is vital for navigating a complex world”. Later in the abstract, “supporting feature-based….” Should probably be “we demonstrate that supporting…” or “we show that supporting…”. In general I think some more brevity and directness would benefit the manuscript.

2. At times it is unclear exactly what version of the model is being discussed, for example, I was confused if OI is disabled for the final paragraph of page 11

3. When referring to “behavioral performance”, it is not always clear what metric you are referring to. …”had a marked behavioral effect – it resulted in a decrease in the inferred probability of the context following the cue on correct trials” is not a behavioral measure. How exactly does this inability to contextualize translate to the actions of the agent, and how is that difference quantified?

4. What happens when OI is left on (after the first 200 trials)?

5. What does “difference in splitter probabilities” mean in Figure 6C? Is this just the fraction of splitters?

Reviewer #2: Summary

Passlack and MacAskill’s study explores how the brain performs contextual inference - the ability to distinguish between hidden contexts and adjust behavior accordingly. The authors focus on two computational strategies: a feature inference (FI) model, which relies on environmental cues, and an outcome inference (OI) model, which relies on behavioral outcomes, and test the performance of these models in differentiating contexts during learning.

The authors propose a joint inference model that combines both strategies during learning. This hybrid approach allows agents to form distinct contextual representations and improves performance in tasks where contextual signals are weak or noisy. The model is tested on several behavioral paradigms, including a cued T-maze, delayed non-match to sample, and structured paired-associates tasks. In each case, the joint model outperforms individual strategies, especially when contexts are similar or overlapping.

The study draws comparisons with neuroscience findings: the joint model reproduces patterns of hippocampal activity known as splitter cells, which fire differently depending on context. Removing outcome inference during learning mimics (somewhat) the loss of these cells observed in experiments where prefrontal input to the hippocampus is disrupted. The authors conclude that outcome-based signals from the prefrontal cortex may support the hippocampus in forming stable, context-specific representations.

Overall, the manuscript is well written and clearly visualized. The model offers a novel computational framework that explains how the brain might integrate environmental features and behavioral outcomes to perform flexible, context-dependent learning. It largely aligns with experimental data and provides insights into the neural mechanisms underlying contextual inference.

Major comments

1. The manuscript aims to address the problem of contextual inference, which typically involves identifying hidden task states based on behavioral or environmental variables. However, in the T-maze paradigm used here, the context appears to be explicitly signaled to the agent via a visible cue. This raises a concern about whether the task truly requires inference. For example, hippocampal place cells are known to remap in response to sensory changes such as alterations in environmental color (e.g., Anderson and Jeffery, 2003; PMID: 14523083), but this reflects a response to overt sensory input rather than inference of a hidden state. Could the authors clarify how their model captures the process of inferring latent context, given that the agent is provided with an explicit cue that directly indicates the current context?

2. In a cued continuous T-maze, Catanese et al. (2012; PMID: 22535656) dissociated splitter cell activity and behavioral choice: when the cue was presented late on the central stem (but still early enough for the animal to make the correct choice) splitter activity continued to reflect the previously learned trajectory until after the decision had been made. A similar absence of splitter activity was reported by Berke et al. (2009; PMID: 19144741) in a cued plus maze. One interpretation of these findings is that cue-guided behavior may not depend on hippocampal processing. If this is the case, it raises questions about the authors’ claim that feature inference (FI) depends on the hippocampus and undermines their observation of splitter cell activity. Could the authors comment on how their model accounts for these findings, and whether FI in their framework necessarily reflects hippocampal involvement in cue-driven behavior?

Minor comments

1. Please include line/page numbers for ease of corrections.

2. Given the discontinuous nature of the T-maze design used in the study, it is not possible to distinguish between retrospective and prospective splitter cell activity. However, Ito et al. (2015; PMID: 26017312) showed that lesions to the nucleus reuniens specifically abolished prospective splitter signals in CA1. Can the authors comment on whether the reduced splitter activity observed in the FI model, relative to the joint model, reflects a retrospective or prospective signal, or whether this distinction can be inferred or speculated upon within their framework?

3. It is well established that hippocampal place cells can remap, and animals can respond to contextual changes, even in the absence of a structured task, for example, during free foraging (Anderson and Jeffery, 2003; PMID: 14523083). Does the author’s model require the presence of task structure to function, or could it also account for more general forms of remapping?

4. Can the authors offer specific experimental predictions based on their model? For instance, in a task designed to dissociate FI and OI contributions, inactivation of the prefrontal cortex selectively impair performance on components relying on outcome inference? Specific, testable predictions would greatly strengthen the value of the model to the experimental community.

5. “Therefore in this instance, past outcome is not informative for behaviour, and so we reasoned that OI agents would not be ale to perform this version of the task.” Typo in ‘able’.

Reviewer #3: General assessment

This manuscript proposes a computational framework combining a feature inference (FI) and an outcome inference (OI) module to account for context-dependent learning in a cued T-maze task. The model aims to capture hippocampal (FI) and prefrontal (OI) contributions to contextual inference. While technically competent and clearly written, the biological motivation and interpretability remain limited. The behavioral paradigm is not clearly tied to a hippocampal-dependent task, and the computational novelty relative to existing latent-state or cue-based inference models is modest.

Overall, the work presents an interesting algorithmic idea—joint inference of context from features and outcomes - but it does not convincingly demonstrate what we learn about hippocampus, prefrontal cortex, or specific behavioral phenomena. Strengthening the biological rationale, clarifying the model’s mechanisms, and situating it more explicitly within existing literature on cue-based learning, contextual inference, and hippocampal representations would substantially improve the manuscript.

1. Major comments

The link between the proposed modules and hippocampal–prefrontal circuitry is tenuous.

The involvement of the hippocampus in the task modeled—a 2-arm maze where an early cue indicates which arm is rewarded - needs to be specified (e.g. Kim, Lee & Lee, 2012; Ainge et al., 2007). Many radial- and T-maze behaviors of this type rely more on anterior cingulate or medial prefrontal regions (Seamans, Floresco & Phillips, 1995). Consequently, the mapping of FI → hippocampus and OI → PFC requires some unpacking.

If Kim et al 2012 is taken as the closest empirical precedent, hippocampus is required only when the cue disappears before the choice, implying a memory or contextual maintenance process. In contrast, the model updates its inferred context continuously at each state and makes the sole decision at the final junction, assuming ongoing re-inference that lacks a clear biological correlate. These assumptions—decision timing and continuous context updating—should be acknowledged explicitly, as they constrain the biological validity of the framework.

More generally, the paper does not clearly show what new insight it provides about hippocampal or prefrontal function. The proposed combination of successor-representation-like inference and outcome-based contextual labeling reads more as a generic algorithmic proposal than a testable neural hypothesis.

2. Model exposition and transparency

The presentation could be streamlined. Clarifying the mechanism under study would improve readability and help the reader focus on the key computational idea.

• Explain clearly how OI and FI are combined for decision-making: both modules infer a context posterior, these are averaged during early trials, and the resulting posterior selects the context-specific TD map for action selection.

• Explicitly define the feature vector used by FI (positions + cue channels, with cue over-representation) and the state representation used by OI (tabular positions).

• Include the essential inference and TD equations in the Results section rather than only in Methods.

3. Biological grounding of the task

The choice of task needs stronger justification. Several studies have demonstrated cue → response T-maze learning without block pre-training (Barnes et al., 2011; Kim et al., 2012; Arlt et al., 2022). The manuscript currently over-states the absence of models for “context-dependent learning without externally provided context signals.”

Here, “context” corresponds to which side of the maze is rewarded - essentially a cued two-alternative-choice situation. The authors should situate their work within the extensive literature on cue-based decisions, latent-state inference (e.g. IBL), and classic context-learning models in spatial navigation (e.g. Tolman’s hidden-state frameworks).

4. Clarify the OI module’s role

The contribution of the OI component is described inconsistently. In the model, the OI posterior is averaged with the FI posterior during the first 200 trials; there is no explicit dynamic stabilization signal from OI to FI. It should be stated clearly that the joint posterior provides cleaner early context labels, allowing FI to learn distinct maps, after which FI operates independently.

The biological analogy (PFC providing a stabilizing signal to HPC) could then be discussed as a supervised labeling phase. The authors should also explain why OI input is required continuously in later tasks but only during learning in the T-maze.

5. Model evaluation and interpretation

• “Training” vs “testing.” These terms are potentially misleading since the “testing” phase still includes reward feedback; only the trial structure changes (random vs blocked). Clarify terminology for consistency.

• “Predictive strategy.” This phrasing is ambiguous. FI performs cue → response mapping within a trial, not across trials. Clarify that “predictive” refers to within-trial cue-based prediction, not cross-trial latent inference.

• Quantitative analysis. It would help to show the relative contributions of FI and OI to the joint posterior and how each fails when used alone. Visualizing the separate context maps MzMz and CzCz would make this clearer.

6. Relation to existing algorithms

The reference to Allen et al. (2024) is intriguing but underdeveloped. Without implementing such automatic arbitration, it would still be informative to analyze how discrepancies between FI and OI posteriors evolve and whether weighting by these discrepancies could explain the observed performance improvements.

7. Claims and literature positioning

At several points the text claims that no algorithms can perform context-dependent learning without an externally provided context signal. This is inaccurate: latent-state and block-prior inference frameworks already address this (e.g. IBL; Gershman 2017 Curr Opin Neurobiol). The authors should soften this statement and acknowledge prior work.

8. “Splitter-cell” analogy

The analogy to hippocampal splitter cells is overstated. Biological splitters encode future trajectory even in cue-free zones (e.g. Ferbinteanu & Shapiro 2003). In contrast, the model’s “splitter-like” activity arises from cue/context features explicitly encoded in the state vector—better described as context selectivity than prospective splitting.

To strengthen this claim:

• Perform position × context decoding or future-turn-conditioned analyses on cue-absent states.

• Clarify how the metric distinguishes “splitter” from “cue” cells.

Minor comments

• Define “attempts on block switches.” Specify how this is computed (e.g. number of incorrect trials after a switch until first correct).

• Clarify Fig. 2C (y-axis label, caption, and interpretation).

• Correct Fig. 5 caption (panel “e” mismatch) and explain how cue representations were computed.

• Add statistical notation in figure legends (* p < 0.05) and refer to Table 6 for detailed tests.

• Either justify or remove the speculative “Lévy flight” comment, which is difficult to interpret in a discrete corridor.

Overall recommendation

The paper presents a creative idea about combining feature- and outcome-based context inference, but the biological interpretation and contextual grounding remain weak. Major revisions clarifying the task relevance, algorithmic transparency, and relationship to existing latent-state models are necessary before the work can make a clear conceptual contribution.

**Have the authors made all data and (if applicable) computational code underlying the findings in their manuscript fully available?**

The PLOS Data policy requires authors to make all data and code underlying the findings described in their manuscript fully available without restriction, with rare exception (please refer to the Data Availability Statement in the manuscript PDF file). The data and code should be provided as part of the manuscript or its supporting information, or deposited to a public repository. For example, in addition to summary statistics, the data points behind means, medians and variance measures should be available. If there are restrictions on publicly sharing data or code —e.g. participant privacy or use of data from a third party—those must be specified.requires authors to make all data and code underlying the findings described in their manuscript fully available without restriction, with rare exception (please refer to the Data Availability Statement in the manuscript PDF file). The data and code should be provided as part of the manuscript or its supporting information, or deposited to a public repository. For example, in addition to summary statistics, the data points behind means, medians and variance measures should be available. If there are restrictions on publicly sharing data or code —e.g. participant privacy or use of data from a third party—those must be specified.

Reviewer #1: Yes

Reviewer #2: Yes

Reviewer #3: None

PLOS authors have the option to publish the peer review history of their article (what does this mean?. If published, this will include your full peer review and any attached files.). If published, this will include your full peer review and any attached files.

**Do you want your identity to be public for this peer review?** For information about this choice, including consent withdrawal, please see our For information about this choice, including consent withdrawal, please see our Privacy Policy ..

Reviewer #1: No

Reviewer #2: No

Reviewer #3: No

**Figure resubmission:**

After uploading your figures to PLOS’s NAAS tool - https://ngplosjournals.pagemajik.ai/artanalysis NAAS will process the files provided and display the results in the "Uploaded Files" section of the page as the processing is complete. If the uploaded figures meet our requirements (or NAAS is able to fix the files to meet our requirements), the figure will be marked as "fixed" above. If NAAS is unable to fix the files, a red "failed" label will appear above. When NAAS has confirmed that the figure files meet our requirements, please download the file via the download option, and include these NAAS processed figure files when submitting your revised manuscript.
---

## [Decision Letter · Decision Letter 1]

24 Feb 2026

Contextual inference through flexible integration of environmental features and behavioural outcomes

PLOS Computational Biology

Dear Dr. MacAskill,

Thank you for submitting your manuscript to PLOS Computational Biology. After careful consideration, we feel that it has merit but does not fully meet PLOS Computational Biology's publication criteria as it currently stands. Therefore, we invite you to submit a revised version of the manuscript that addresses the points raised during the review process.

We look forward to receiving your revised manuscript.

Kind regards,

Alejandro Tabas, Ph.D.

Academic Editor

PLOS Computational Biology

Daniele Marinazzo

Section Editor

PLOS Computational Biology

**Additional Editor Comments (if provided):**

**Journal Requirements:**

**Reviewers' comments:**

Reviewer's Responses to Questions

**Comments to the Authors:**

Reviewer #1: The authors have fully addressed our concerns.

Reviewer #2: The authors have addressed my concerns and greatly improved the manuscript. I would recommend publication in PLOS Comp Bio.

Reviewer #3: I thank and congratulate the authors for their careful and detailed response to my previous review and for the substantial effort invested in revising the manuscript. Many of the points I raised in the first round have been thoughtfully addressed, and the manuscript is clearer in several respects, particularly in the description of the model components and the expanded discussion. I appreciate the authors’ engagement with the critiques and the clarifications they have added.

Below I outline a set of remaining points that, in my view, would further strengthen the manuscript. These comments are intended to help sharpen both the biological interpretation and the presentation of the proposed framework.

Major comments

Biological interpretation and task relevance

(Lines 430, 463–467, 592, 700, 827, 1025, 1059, 1080, 1120–1122)

While the model is internally consistent, the biological interpretation remains challenging. The cued two-arm T-maze used here is not unambiguously hippocampal-dependent, and prior experimental work suggests that reliance on hippocampus versus mPFC/ACC depends strongly on task timing, cue availability, and memory demands (e.g., Kim et al., 2012; Ainge et al., 2007; Seamans et al., 1995). It would strengthen the manuscript to more explicitly delineate which aspects of the modeled behavior are expected to rely on hippocampus versus prefrontal cortex, and under what task constraints.

Relatedly, several claims regarding context-specific representations (e.g., distinct maps, splitter-like activity) would benefit from a clearer distinction between context selectivity driven by explicit cue features and prospective trajectory coding as reported in hippocampal physiology. As currently phrased, some statements may read as broader than what the task and model can directly support.

Mechanistic clarity of FI–OI interaction

(Lines 430, 649, 700, 722, 827, 939)

The description of OI “supporting” FI would benefit from additional precision. In the current implementation, OI contributes via averaging of posteriors during early learning, rather than via an ongoing stabilizing or modulatory signal. Explicitly framing this interaction as an early supervised labeling phase -after which FI operates largely autonomously - would help clarify both the computational mechanism and the proposed biological analogy.

In addition, it would be helpful to explain why OI input appears to be required continuously in later tasks but only during learning in the T-maze, and what this implies about the generality or limits of the proposed framework.

Inference dynamics, decision timing, and trajectory control

(Lines 795, 1025, 1059, 1080)

The manuscript would benefit from greater clarity regarding when decisions are assumed to occur. While the model updates context continuously as the agent progresses through states, behaviorally the task may permit an early commitment upon cue presentation, followed by execution rather than ongoing inference. Clarifying whether decisions are assumed to be made at cue onset, at the junction, or continuously would help align the model with existing decision-making frameworks (e.g., hierarchical or option-based control) and avoid over-interpreting sequential inference in this task.

Comparison to existing models

(Lines 430, 467; Line 1161)

Several statements suggest that the proposed framework addresses limitations of existing contextual inference algorithms. It would strengthen the contribution to more explicitly compare the present approach to latent-state, block-prior, and hierarchical reinforcement-learning models, clarifying what is gained - and what remains limited - by the FI–OI combination relative to this existing literature.

Relatedly, statements such as “to confirm that Bayesian inference is necessary…” (Lines 466–467) may read as overly general. It may be more accurate to state that Bayesian inference is necessary within the present modeling framework or set of comparisons, rather than implying necessity for contextual inference more broadly.

Quantitative and experimental grounding

(Lines 463–467, 487, 649, 722)

Where quantitative performance metrics are reported (e.g., number of attempts on block switches), comparison to experimental learning rates or behavioral variability would help contextualize the results. Even a qualitative discussion of how these values relate to rodent performance would improve biological interpretability.

Minor comments

Feature representation and architectural assumptions

(Line 352; Line 430)

The scaling of cue features (e.g., factor of 4) appears important for model performance. It would be helpful to state explicitly whether this scaling is necessary for FI to function robustly and how it should be interpreted biologically. Relatedly, clarifying why OI is restricted from accessing cue information (architectural assumption vs biological motivation) would improve transparency.

Interpretation of results versus model design

(Line 495)

The statement that FI agents “formed distinct successor maps” appears largely driven by model design. Reframing this as a mechanism that enables separation—rather than as an empirical discovery—would improve precision.

Successor representation flexibility

(Line 1157)

The discussion of SR flexibility appears to conflate reward revaluation with transition structure changes. Since classical SR is not flexible to changes in transition dynamics without recomputation, moderating this claim would improve technical accuracy.

Inference dynamics and observability

(Lines 1209, 1296)

Inference in the model occurs continuously at each state, whereas biologically inference is often event-triggered (e.g., at cue onset or outcome). Acknowledging this distinction would strengthen the discussion. Similarly, the cue is fully informative but temporally displaced from action, which may be more accurately described as delayed information rather than partial observability.

Presentation and clarity

(Lines 518, 529, 722)

Several minor presentation issues remain, including typographical corrections, clarification of terminology, and moderation of some over-general statements. The conclusion and discussion sections could likely be shortened to improve focus and avoid over-statement.

Overall, I appreciate the authors’ efforts to address the initial concerns and believe the manuscript has improved. Further clarifying the biological interpretation, tightening the link to existing experimental work, and sharpening the mechanistic claims would significantly strengthen the contribution.

**Have the authors made all data and (if applicable) computational code underlying the findings in their manuscript fully available?**

The PLOS Data policy requires authors to make all data and code underlying the findings described in their manuscript fully available without restriction, with rare exception (please refer to the Data Availability Statement in the manuscript PDF file). The data and code should be provided as part of the manuscript or its supporting information, or deposited to a public repository. For example, in addition to summary statistics, the data points behind means, medians and variance measures should be available. If there are restrictions on publicly sharing data or code —e.g. participant privacy or use of data from a third party—those must be specified.requires authors to make all data and code underlying the findings described in their manuscript fully available without restriction, with rare exception (please refer to the Data Availability Statement in the manuscript PDF file). The data and code should be provided as part of the manuscript or its supporting information, or deposited to a public repository. For example, in addition to summary statistics, the data points behind means, medians and variance measures should be available. If there are restrictions on publicly sharing data or code —e.g. participant privacy or use of data from a third party—those must be specified.

Reviewer #1: Yes

Reviewer #2: Yes

Reviewer #3: None

PLOS authors have the option to publish the peer review history of their article (what does this mean?. If published, this will include your full peer review and any attached files.). If published, this will include your full peer review and any attached files.

**Do you want your identity to be public for this peer review?** For information about this choice, including consent withdrawal, please see our For information about this choice, including consent withdrawal, please see our Privacy Policy ..

Reviewer #1: **Yes:** Rui Ponte CostaRui Ponte Costa

Reviewer #2: No

Reviewer #3: No

**Figure resubmission:**

**Reproducibility:**



---

## [Editor Report · Decision Letter 2]

5 Mar 2026

Dear Prof MacAskill,

We are pleased to inform you that your manuscript 'Contextual inference through flexible integration of environmental features and behavioural outcomes' has been provisionally accepted for publication in PLOS Computational Biology.

Best regards,

Alejandro Tabas, Ph.D.

Academic Editor

PLOS Computational Biology

Daniele Marinazzo

Section Editor

PLOS Computational Biology
